



# Prognostic Assumed-PDF (DDF) Approach: Further Generalization and Demonstrations

Yano Jun-Ichi Yano

CNRM UMR3589 (CNRS), Météo France

**Correspondence:** Jun-Ichi Yano (jun-ichi.yano@cnrs.fr)

**Abstract.**

A methodology for directly predicting the time evolution of the assumed parameters for the distribution densities based on
the Liouville equation, as proposed earlier, is extended to multi–dimensional cases as well as when the systems are constrained by integrals over a part of the variable range. The extended methodology is tested against a convective energy cycle system as well as the Lorenz's stranger attractor. As a general tendency, the variance tends to collapse to a vanishing value over a finite time regardless of the chosen assumed distribution form. This general tendency is likely due to the common cause as collapse of the variance commonly found in ensemble–based data assimilation.

## 1  Introduction

One of important manners to characterize the nonlinear systems is by predicting evolution of the distributions of the variables in space as well as in values themselves, but also as a probability. The important applications in geophysics include the subgrid–scale parameterizations based on the distribution density functions (DDFs: Sommeria and Deadorff 1977, Mellor 1977, Bougeault 1981, LeTreut and Li 1991, Bechtold *et al.* 1992, 1995, Richard and Royer 1993, Bony and Emanuel 2001,
Golaz *et al.* 2002, Tompkins 2002), the particle–size distributions in cloud microphysics (*cf*., Khain *et al.* 2014, Khain and Pinsky 2018), and the characterizations of data and model uncertainties by the probability density functions (PDFs) in data assimilations (Carrassi *et al.* 2018, Evensen *et al.* 2022).

Yano, Larson, and Phillips (2024, YLP) have proposed a general methodology for evaluating the evolution of those distributions in efficient manner. The essence of their approach may be called the prognostic assumed–PDF (DDF), as an extension
of the classical assumed PDF approaches (*cf*., Golaz *et al.* 2002). In YLP, only one–variable cases under relatively simple constraints (*i.e.*, output conditions) have been considered. The purpose of the present study is to further generalize this methodology to the multi-dimensional systems with more general constraints, and to present further demonstrative cases.





The presentation will be developed in the following manner. Sec. 2 briefly summarizes the formulation presented in YLP, and generalizes it into the cases 1) that constraints are defined over limited integral ranges, and 2) when different assumed distribution forms are assumed over the different domains. Sec. 2 first generalizes those one–variable results for the assumed–PDF formulation into the multi–variable systems in a general manner, without specifying a PDF form. Sec. 3 shows in a more concrete manner, how these general formulations can be used when a Gaussian distribution with two variables are assumed. This example also suggests how reductions with different assumed–PDF forms can be proceeded. The general formulations presented in Secs. 2 and 3 are applied to the two dynamical systems in Secs. 4 and 5: (i) the two–variable convective energy–cycle system introduced by Yano and Plant (2012, Sec. 4), and (ii) the three–variable system of Lorenz's strange attractor (Sec. 5). Three different assumed–PDF forms are considered for each system, and results are discussed. Sec. 6 summarizes the results with further discussions.

## 2 General Formulation With One Variable

### 2.1 Basic Formulation

As in YLP, for now, let us assume a dynamical system with a single variable, $\phi$, and that a distribution of $\phi$ can be approximately represented by an assumed form,

$$p = p(\phi; \lambda_0, \lambda_1, \ldots, \lambda_N), \tag{2.1.1}$$

which is characterized by $N$ parameters, $\lambda_j$ $(j = 1, \ldots, N)$. We also separately introduce a normalization factor, $\lambda_0$, that satisfies a relation of $p \propto \lambda_0$. It follows that

$$\frac{\partial p}{\partial \lambda_0} = \frac{p}{\lambda_0}. \tag{2.1.2}$$

The distribution Eq. (2.1.1) is normalized by

$$\int p \, d\phi = 1. \tag{2.1.3}$$

Here and in the following, an unspecified integral range may be taken from $-\infty$ to $+\infty$ with many of the physical variables, but some physical variables are semi–positive definite (*e.g.*, temperature, mixing ratios). In the latter case, the integral range above must be from 0 to $+\infty$.

Eovlution of this distribution, $p$, is governed by the Liouville equation

$$\frac{\partial p}{\partial t} = -\frac{\partial pS}{\partial \phi}, \tag{2.1.4}$$

when the dynamical system is defined by $\dot{\phi} = S$. From Eq. (2.1.3),

$$\frac{\dot{\lambda}_0}{\lambda_0} = -\sum_{i=1}^{N} [\int \frac{\partial p}{\partial \lambda_i} d\phi] \dot{\lambda}_i. \tag{2.1.5}$$



Inserting Eq. (2.1.1) into Eq. (2.1.4), weighting it by $\sigma_l$ ($l = 1, \ldots, N$), and integrating it over the full variable range, we obtain a final expression for the prognostic equation for the distribution parameters, $\{\lambda_j\}$:

$$\sum_{j=1}^{N} \dot{\lambda}_j [\int \sigma_l \frac{\partial p}{\partial \lambda_j} d\phi - \int \sigma_l p \, d\phi \int \frac{\partial p}{\partial \lambda_j} d\phi] = \int p S \frac{\partial \sigma_l}{\partial \phi} d\phi \tag{2.1.6}$$

for $l = 1, \ldots, N$. We can see from Eq. (2.1.6) that the weights, $\sigma_l$, are most conveniently chosen in such a manner that

$$\langle \sigma_l \rangle = \int p \sigma_l d\phi, \quad l = 1, \ldots, N \tag{2.1.7}$$

constitute the constraints for this distribution: YLP suggest to choose those constraints to be the outputs that are required in a host model. Since the left–hand side is equal to $d\langle \sigma_l \rangle / dt$, Eq. (2.1.6) can predict those constraints in self–consistent manner under an assumed form Eq. (2.1.1).

     In the following two subsections, this basic formulation is generalized into the two manners: first, the constraints are generalized to the cases that integral ranges are only over limited ranges of the distribution variable, $\phi$; second, when the distributions

take different forms over those limited ranges, as a consequence.

## 2.2    When the constraints are defined over limited integral ranges

Now, we generalize the above basic formulation to the cases that the different constraints are introduced over two different ranges of the distribution variables. More specifically, we assume that the distribution variable, $\phi$, is defined over $[-\infty, +\infty]$, and the constraints are introduced in analogy with Eq. (2.1.6), but differently over the two ranges, $[-\infty, 0]$ and $[0, +\infty]$:

$$\int_{0}^{+\infty} F_l^+(p, \phi) d\phi = C_l^+, \quad l = 1, \ldots, N^+ \tag{2.2.1a}$$

$$\int_{-\infty}^{0} F_l^-(p, \phi) d\phi = C_l^-, \quad l = 1, \ldots, N^- \tag{2.2.1b}$$

In this case, the variational principle to maximize the information entropy with the given constraints is given by:

$$\delta[-\int_{-\infty}^{+\infty} p \log p \, d\phi - \sum_{l=0}^{N^+} \lambda_l^{+\prime} \int_{0}^{+\infty} F_l^+(p, \phi) d\phi - \sum_{l=0}^{N^-} \lambda_l^{-\prime} \int_{-\infty}^{0} F_l^-(p, \phi) d\phi] = 0. \tag{2.2.2}$$

It reduces to:

$$\int_{0}^{+\infty} (\log p + \sum_{l=0}^{N^+} \lambda_l^+ \frac{\partial F_l^+}{\partial p}) \delta p \, d\phi + \int_{-\infty}^{0} (\log p + \sum_{l=0}^{N^-} \lambda_l^+ \frac{\partial F_l^-}{\partial p}) \delta p \, d\phi = 0. \tag{2.2.3 incorrect}$$

Thus, the most–likley distribution under the constraints (2.2.1a, b) is:

$$p = \begin{cases} p_0^+ \exp[-\sum_{l=1}^{N^+} \lambda_l^+ \frac{\partial F_l^+}{\partial p}], & \phi > 0 \\[2mm] p_0^- \exp[-\sum_{l=1}^{N^-} \lambda_l^- \frac{\partial F_l^-}{\partial p}], & \phi < 0. \end{cases} \tag{2.2.3}$$





Note that $p_0^+ = p_0^-$, because at $\phi = 0$, $p^+ = p^-$.

### 2.3 When the distribution takes different forms in two different domains

In the last subsection, the constraints have been generalized to a case that those are defined over two limited ranges of the distribution variable, $\phi$: $cf.$, Eqs. (2.2.1a, b). As it turns out, the most–likely distribution (Eq. 2.2.3) under these constraints also take different forms over those two ranges. Consequently, the formulation for predicting the given PDF parameters must also be generalized to such cases: this subsection addresses this issue.

By following the last subsection, we divide the variable range into the two subdomains, $[-\infty, 0]$ and $[0, +\infty]$, and assume
that the distribution takes different forms over those two subdomains. Thus:

$$p = \begin{cases} p^+(\phi, \lambda_0^+, \lambda_1^+, \ldots, \lambda_N^+), & \phi > 0 \\ p^-(\phi, \lambda_0^-, \lambda_1^-, \ldots, \lambda_N^-), & \phi < 0, \end{cases} \tag{2.3.1}$$

assuming $N^+ = N^- = N$ for now. By continuity at $\phi = 0$, $p^+ = p^-$. As before, we assume that the first parameters, $\lambda_0^\pm$, are the normalization factors, thus:

$$\frac{\partial p}{\partial \lambda_0^+} = \frac{p}{\lambda_0^+}, \phi > 0 \tag{2.3.2a}$$

$$\frac{\partial p}{\partial \lambda_0^-} = \frac{p}{\lambda_0^-}, \phi < 0, \tag{2.3.2b}$$

Especially when $p|_{\phi = \pm 0} = \lambda_0^\pm$, as the case with the result (2.2.3) it follows that $\lambda_0^+ = \lambda_0^- (= \lambda_0)$. Here, the assumed form (2.3.1) follows from the constraints of the form (2.2.1a, b).

By applying the time derivative to the normalization condition,

$$\int\limits_{-\infty}^{+\infty} p \, d\phi = 1,$$

we find:

$$\frac{\partial}{\partial t} \int\limits_0^{+\infty} p \, d\phi + \frac{\partial}{\partial t} \int\limits_{-\infty}^0 p \, d\phi = 0, \tag{2.3.3}$$

which reduces to:

$$\frac{\dot{\lambda}_0}{\lambda_0} = -\sum_{i=1}^N [\dot{\lambda}^+{}_i \int\limits_0^{+\infty} \frac{\partial p}{\partial \lambda_i^+} d\phi + \dot{\lambda}^-{}_i \int\limits_{-\infty}^0 \frac{\partial p}{\partial \lambda_i^-} d\phi] \tag{2.3.4}$$

noting that $\lambda_0^+ = \lambda_0^- \equiv \lambda_0$ and $p_+ + p_- = 1$, where

$$p_+ = \int\limits_0^{+\infty} p \, d\phi, \quad p_- = \int\limits_{-\infty}^0 p \, d\phi$$





After substituting the assumed PDF form (2.3.1), and applying the chain rules on the time derivative in the Liouville equation (2.1.4), it reduces to:

$$p\frac{\dot{\lambda}_0}{\lambda_0} + \sum_{i=1}^{N} \frac{\partial p}{\partial \lambda_i^+} \dot{\lambda}_i^+ + \frac{\partial}{\partial \phi}(pS) = 0, \; \phi > 0 \tag{2.3.5a}$$

$$p\frac{\dot{\lambda}_0}{\lambda_0} + \sum_{i=1}^{N} \frac{\partial p}{\partial \lambda_i^-} \dot{\lambda}_i^- + \frac{\partial}{\partial \phi}(pS) = 0, \; \phi < 0. \tag{2.3.5b}$$

By applying weighted intergrals on both, and further substituting Eq. (2.3.4) into the above, we obtain a pair of equations for predicting the PDF parameters:

$$\sum_{i=1}^{N}\{\dot{\lambda}_i^+ [\int_0^{+\infty} \sigma_l^+ \frac{\partial p}{\partial \lambda_i^+} d\phi - \int_0^{+\infty} \sigma_l^+ pd\phi \int_0^{+\infty} \frac{\partial p}{\partial \lambda_i^+} d\phi] - \dot{\lambda}_i^- \int_0^{+\infty} \sigma_l^+ pd\phi \int_{-\infty}^0 \frac{\partial p}{\partial \lambda_i^-} d\phi\} + \int_0^{+\infty} \sigma_l^+ \frac{\partial}{\partial \phi}(pS)d\phi = 0, \tag{2.3.6a}$$

$$\sum_{i=1}^{N}\{-\dot{\lambda}_i^+ \int_{-\infty}^0 \sigma_l^- pd\phi \int_0^{+\infty} \frac{\partial p}{\partial \lambda_i^+} d\phi + \dot{\lambda}_i^- [\int_{-\infty}^0 \sigma_l^- \frac{\partial p}{\partial \lambda_i^-} d\phi - \int_{-\infty}^0 \sigma_l^- pd\phi \int_{-\infty}^0 \frac{\partial p}{\partial \lambda_i^-} d\phi]\} + \int_{-\infty}^0 \sigma_l^- \frac{\partial}{\partial \phi}(pS)d\phi = 0 \tag{2.3.6b}$$

for $l = 1, \dots, N$. Here, $\sigma_l^{\pm} = F_l^{\pm}/p_{\pm}$ by following the output–constrained distribution principle proposed in YLP.

## 2.4 Multidimensional Case (I)

The formulations introduced over the last subsections are now further generalized into the multidimensional case, in which the distribution depends on more than one variable. This variable set is treated as a vector, $\mathbf{x}$, with the first component corresponds to $x$. Here, note a change of the notation for the distribution variable from $\phi$ to $\mathbf{x}$ from now on. As in the last two subsections, the distribution is defined differently depending on the sign of $x$:

$$p = \begin{cases} p^+(\mathbf{x}, \lambda_0^+, \lambda_1^+, \dots, \lambda_N^+, \lambda_{N+1}, \dots, \lambda_{N+M}), & x > 0 \\ p^-(\mathbf{x}, \lambda_0^-, \lambda_1^-, \dots, \lambda_N^-, \lambda_{N+1}, \dots, \lambda_{N+M}) & x < 0 \end{cases}$$

Here, the first $N$ parameters, $\{\lambda_l^{\pm}\}$ ($l = 1, \dots, N$), take different definitions for positive and negative sides of $x$, whereas the last $M$ parameters, $\{\lambda_{N+l}\}$ ($l = 1, \dots, M$), are assumed to be common. This assumed form results when there are $N$ constraints defined differently depending on the sign of $x$, as in the form of (2.2.1a, b), and another $M$ constraints that do not depend on the sign of $x$. Furthermore, by following the output–constrained distribution principle proposed in YLP, the weights to be introduced are: $\sigma_l^{\pm} = F_l^{\pm}/p_{\pm}$ for $l = 1, \dots, N$, and $\sigma_l = F_l/p$ for $l = N+1, \dots, N+M$, by following the notations in Eqs. (2.2.1a, b).





However, we drop the superscript $\pm$ on $p$ for now for simplicity. Thus, repeating the same procedure as the last subsection, we obtain a pair of prognostic equations for the PDF parameters:

$$\sum_{i=1}^{N}\{\dot{\lambda}_i^+[\int_+ \sigma_l \frac{\partial p}{\partial \lambda_i^+}d\mathbf{x} - \int_+ \sigma_l p d\mathbf{x}\int_+ \frac{\partial p}{\partial \lambda_i^+}d\mathbf{x}] - \dot{\lambda}_i^- \int_+ \sigma_l p d\mathbf{x}\int_- \frac{\partial p}{\partial \lambda_i^-}d\mathbf{x}\}$$

$$+\sum_{i=1}^{M}\dot{\lambda}_{N+i}[\int_+ \sigma_l \frac{\partial p}{\partial \lambda_{N+i}}d\mathbf{x} - \int_+ \sigma_l p d\mathbf{x}\int_+ \frac{\partial p}{\partial \lambda_{N+i}}d\mathbf{x}] + \int_+ \sigma_l \frac{\partial}{\partial x}(pS)d\mathbf{x} = 0, \tag{2.4.1a}$$

$$\sum_{i=1}^{N}\{-\dot{\lambda}_i^+ \int_- \sigma_l p d\mathbf{x}\int_+ \frac{\partial p}{\partial \lambda_i^+}d\mathbf{x} + \dot{\lambda}_i^- [\int_- \sigma_l \frac{\partial p}{\partial \lambda_i^-}d\mathbf{x} - \int_- \sigma_l p d\mathbf{x}\int_- \frac{\partial p}{\partial \lambda_i^-}d\mathbf{x}]\}$$

$$+\sum_{i=1}^{M}\dot{\lambda}_{N+i}[\int_- \sigma_l \frac{\partial p}{\partial \lambda_{N+i}}d\mathbf{x} - \int_- \sigma_l p d\mathbf{x}\int_- \frac{\partial p}{\partial \lambda_{N+i}}d\mathbf{x}] + \int_- \sigma_l \frac{\partial}{\partial x}(pS)d\mathbf{x} = 0 \tag{2.4.1b}$$

where $\int_\pm$ suggest integrals over $x > 0$ and $x < 0$, respectively, depending on the sign.

Taking the sum of the two, we obtain:

$$\sum_{i=1}^{N}\{\dot{\lambda}_i^+[\int_+ \sigma_l \frac{\partial p}{\partial \lambda_i^+}d\mathbf{x} - \int \sigma_l p d\mathbf{x}\int_+ \frac{\partial p}{\partial \lambda_i^+}d\mathbf{x}] + \dot{\lambda}_i^-[\int_- \sigma_l \frac{\partial p}{\partial \lambda_i^-}d\mathbf{x} - \int \sigma_l p d\mathbf{x}\int_- \frac{\partial p}{\partial \lambda_i^-}d\mathbf{x}]\}$$

$$+\sum_{i=1}^{M}\dot{\lambda}_{N+i}[\int \sigma_l \frac{\partial p}{\partial \lambda_{N+i}}d\mathbf{x} - \int \sigma_l p d\mathbf{x}\int \frac{\partial p}{\partial \lambda_{N+i}}d\mathbf{x}] + \int \sigma_l \frac{\partial}{\partial \phi}(pS)d\mathbf{x} = 0 \tag{2.4.1c}$$

Especially when $\sigma_l$ does not depend on the sign of $x$ (*i.e.*, for $l = N+1,\dots,N+M$), and the distribution is separable with those two distributions variables, then the first N–sum disappears in Eq. (2.4.1c), and

$$\sum_{i=1}^{M}\dot{\lambda}_{N+i}[\int \sigma_l \frac{\partial p}{\partial \lambda_{N+i}}d\mathbf{x} - \int \sigma_l p d\mathbf{x}\int \frac{\partial p}{\partial \lambda_{N+i}}d\mathbf{x}] + \int \sigma_l \frac{\partial}{\partial \phi}(pS)d\mathbf{x} = 0. \tag{2.4.1d}$$

Thus, the last $M$ parameters, $\lambda_{N+l}$ ($l = 1,\dots,M$), can be predicted by a single set of equations (2.4.1d). Note also that the prognostic equations for the PDF parameters reduce to Eq. (2.4.1d) when all the constraints are defined over the full range so that $N = 0$.

## 2.5 Multidimensional Case (II)

In this subsection, we further subdevide the domain in $y$ direction, thus:

$$p = \begin{cases} p^{++}(\mathbf{x},\lambda_0^{++},\lambda_1^{++},\dots,\lambda_N^{++},\lambda_{N+1},\dots,\lambda_{N+M}), & x > 0, y > 0 \\ p^{+-}(\mathbf{x},\lambda_0^{+-},\lambda_1^{+-},\dots,\lambda_N^{+-},\lambda_{N+1},\dots,\lambda_{N+M}), & x > 0, y < 0 \\ p^{-+}(\mathbf{x},\lambda_0^{-+},\lambda_1^{-+},\dots,\lambda_N^{-+},\lambda_{N+1},\dots,\lambda_{N+M}) & x < 0, y > 0 \\ p^{--}(\mathbf{x},\lambda_0^{--},\lambda_1^{--},\dots,\lambda_N^{--},\lambda_{N+1},\dots,\lambda_{N+M}) & x < 0, y < 0 \end{cases}$$





where the double superscripts $\pm\pm$ are introduced to indicate the sides in both $x$ and $y$ directions. Otherwise, the conventions for the definitions of the $N + M$ PDF parameters remain the same.

In this case, the weights to be introduced are: $\sigma_l^{\pm\pm} = F_l^{\pm\pm}/p_{\pm\pm}$ for $l = 1,\ldots,N$, and $\sigma_l = F_l/p$ for $l = N+1,\ldots,N+M$, by following similar notations as in Eqs. (2.2.1a, b). Thus, the first $N$ weights take four different definitions depending on the signs of $x$ and $y$. These PDF parameters can be predicted by the equations:

$$
\sum_{i=1}^{N}\{\dot{\lambda}_i^{++}[\int_{++} \sigma \frac{\partial p}{\partial \lambda_i^{++}}d\mathbf{x} - \int_{++} \sigma p d\mathbf{x} \int_{++} \frac{\partial p}{\partial \lambda_i^{++}}d\mathbf{x}]
$$

$$
- \dot{\lambda}_i^{+-}\int_{++} \sigma p d\mathbf{x} \int_{+-} \frac{\partial p}{\partial \lambda_i^{+-}}d\mathbf{x} - \dot{\lambda}_i^{-+}\int_{++} \sigma p d\mathbf{x} \int_{-+} \frac{\partial p}{\partial \lambda_i^{-+}}d\mathbf{x} - \dot{\lambda}_i^{--}\int_{++} \sigma p d\mathbf{x} \int_{--} \frac{\partial p}{\partial \lambda_i^{--}}d\mathbf{x}\}
$$

$$
+ \sum_{i=1}^{M}\dot{\lambda}_{N+i}[\int_{++} \sigma \frac{\partial p}{\partial \lambda_{N+i}}d\mathbf{x} - \int_{++} \sigma p d\mathbf{x} \int \frac{\partial p}{\partial \lambda_{N+i}}d\mathbf{x}] + \int_{++} \sigma \frac{\partial}{\partial \phi}(pS)d\mathbf{x} = 0, \tag{2.4.2a}
$$

$$
\sum_{i=1}^{N}\{-\dot{\lambda}_i^{++}\int_{+-} \sigma p d\mathbf{x} \int_{++} \frac{\partial p}{\partial \lambda_i^{++}}d\mathbf{x} + \dot{\lambda}_i^{+-}[\int_{+-} \sigma \frac{\partial p}{\partial \lambda_i^{+-}}d\mathbf{x} - \int_{+-} \sigma p d\mathbf{x} \int_{+-} \frac{\partial p}{\partial \lambda_i^{+-}}d\mathbf{x}]
$$

$$
- \dot{\lambda}_i^{-+}\int_{+-} \sigma p d\mathbf{x} \int_{-+} \frac{\partial p}{\partial \lambda_i^{-+}}d\mathbf{x} - \dot{\lambda}_i^{--}\int_{+-} \sigma p d\mathbf{x} \int_{--} \frac{\partial p}{\partial \lambda_i^{--}}d\mathbf{x}\}
$$

$$
+ \sum_{i=1}^{M}\dot{\lambda}_{N+i}[\int_{+-} \sigma \frac{\partial p}{\partial \lambda_{N+i}}d\mathbf{x} - \int_{+-} \sigma p d\mathbf{x} \int \frac{\partial p}{\partial \lambda_{N+i}}d\mathbf{x}] + \int_{+-} \sigma \frac{\partial}{\partial \phi}(pS)d\mathbf{x} = 0, \tag{2.4.2b}
$$

$$
\sum_{i=1}^{N}\{-\dot{\lambda}_i^{++}\int_{-+} \sigma p d\mathbf{x} \int_{++} \frac{\partial p}{\partial \lambda_i^{++}}d\mathbf{x} - \dot{\lambda}_i^{+-}\int_{-+} \sigma p d\mathbf{x} \int_{+-} \frac{\partial p}{\partial \lambda_i^{+-}}d\mathbf{x}
$$

$$
+ \dot{\lambda}_i^{-+}[\int_{-+} \sigma \frac{\partial p}{\partial \lambda_i^{-+}}d\mathbf{x} - \int_{-+} \sigma p d\mathbf{x} \int_{-+} \frac{\partial p}{\partial \lambda_i^{-+}}d\mathbf{x}] - \dot{\lambda}_i^{--}\int_{-+} \sigma p d\mathbf{x} \int_{--} \frac{\partial p}{\partial \lambda_i^{--}}d\mathbf{x}\}
$$

$$
+ \sum_{i=1}^{M}\dot{\lambda}_{N+i}[\int_{-+} \sigma \frac{\partial p}{\partial \lambda_{N+i}}d\mathbf{x} - \int_{-+} \sigma p d\mathbf{x} \int \frac{\partial p}{\partial \lambda_{N+i}}d\mathbf{x}] + \int_{-+} \sigma \frac{\partial}{\partial \phi}(pS)d\mathbf{x} = 0, \tag{2.4.2c}
$$

$$
\sum_{i=1}^{N}\{-\dot{\lambda}_i^{++}\int_{--} \sigma p d\mathbf{x} \int_{++} \frac{\partial p}{\partial \lambda_i^{++}}d\mathbf{x} - \dot{\lambda}_i^{+-}\int_{--} \sigma p d\mathbf{x} \int_{+-} \frac{\partial p}{\partial \lambda_i^{+-}}d\mathbf{x} - \dot{\lambda}_i^{-+}\int_{--} \sigma p d\mathbf{x} \int_{-+} \frac{\partial p}{\partial \lambda_i^{-+}}d\mathbf{x}
$$

$$
+ \dot{\lambda}_i^{--}[\int_{--} \sigma \frac{\partial p}{\partial \lambda_i^{--}}d\mathbf{x} - \int_{--} \sigma p d\mathbf{x} \int_{--} \frac{\partial p}{\partial \lambda_i^{--}}d\mathbf{x}]\} + \sum_{i=1}^{M}\dot{\lambda}_{N+i}[\int_{--} \sigma \frac{\partial p}{\partial \lambda_{N+i}}d\mathbf{x} - \int_{--} \sigma p d\mathbf{x} \int \frac{\partial p}{\partial \lambda_{N+i}}d\mathbf{x}] + \int_{--} \sigma \frac{\partial}{\partial \phi}(pS)d\mathbf{x} = 0, \tag{2.4.2d}
$$




By further taking the sum of the four, we again obtain extra constraints:

$$
\sum_{i=1}^{N} \{ \dot{\lambda}_i^+ [ \underset{++}{\int} \sigma \frac{\partial p}{\partial \lambda_i^{++}} d\mathbf{x} - \int \sigma p d\mathbf{x} \underset{++}{\int} \frac{\partial p}{\partial \lambda_i^{++}} d\mathbf{x} ] + [ \underset{+-}{\int} \sigma \frac{\partial p}{\partial \lambda_i^{+-}} d\mathbf{x} - \int \sigma p d\mathbf{x} \underset{+-}{\int} \frac{\partial p}{\partial \lambda_i^{+-}} d\mathbf{x} ]
$$

$$
+ \dot{\lambda}_i^{-+} [ \underset{-+}{\int} \sigma \frac{\partial p}{\partial \lambda_i^{-+}} d\mathbf{x} - \int \sigma p d\mathbf{x} \underset{-+}{\int} \frac{\partial p}{\partial \lambda_i^{-+}} d\mathbf{x} ] + \dot{\lambda}_i^{--} [ \underset{--}{\int} \sigma \frac{\partial p}{\partial \lambda_i^{--}} d\mathbf{x} - \int \sigma p d\mathbf{x} \underset{--}{\int} \frac{\partial p}{\partial \lambda_i^{--}} d\mathbf{x} ] \}
$$

$$
+ \sum_{i=1}^{M} \dot{\lambda}_{N+i} [ \int \sigma \frac{\partial p}{\partial \lambda_{N+i}} d\mathbf{x} - \int \sigma p d\mathbf{x} \int \frac{\partial p}{\partial \lambda_{N+i}} d\mathbf{x} ] + \int \sigma \frac{\partial}{\partial \phi} (pS) d\mathbf{x} = 0 \tag{2.4.2e}
$$

Especially when $\sigma$ does not depend on the direction distinguishing the subscripts $\pm$ in integrals above, the first N–sum disappears in (2.9e) for the separable distributions, and we obtain:

$$
\sum_{i=1}^{M} \dot{\lambda}_{N+i} [ \int \sigma \frac{\partial p}{\partial \lambda_{N+i}} d\mathbf{x} - \int \sigma p d\mathbf{x} \int \frac{\partial p}{\partial \lambda_{N+i}} d\mathbf{x} ] + \int \sigma \frac{\partial}{\partial \phi} (pS) d\mathbf{x} = 0 \tag{2.4.2f}
$$

The formulations in these two last subsections will be applied in more specific cases considered in Secs. 4 and 5.

# 3   Example: Gauss Distribution with Two Variables

As a specific example of multi–dimensional case, in this section, we consider a two–dimensional Gaussian distribution, setting $\mathbf{x} = (x, y)$ and $\mathbf{S} = (S_x, S_y)$:

$$
p = p_0 \exp[-\lambda_1 (x - \bar{x})^2 - \lambda_2 (y - \bar{y})^2 - \lambda_3 (x - \bar{x})(y - \bar{y})]. \tag{3.1}
$$

From the normalization condition,

$$
p_0 = \frac{(\lambda_1 \lambda_2 + \lambda_3^2/4)^{1/2}}{\pi} \tag{3.2}
$$

as derived in the Appendix A.1.

For the distribution (3.1), Eq. (2.4.1d) reduces to:

$$
\sum_{i=1}^{3} [\int \int \sigma \frac{\partial p}{\partial \lambda_i} dx dy - \int \int \sigma p \, dx dy \int \int \frac{\partial p}{\partial \lambda_i} dx dy] \dot{\lambda}_i + \sum_{i=1}^{2} [\int \int \sigma \frac{\partial p}{\partial \bar{x}_i} dx dy - \int \int \sigma p \, dx dy \int \int \frac{\partial p}{\partial \bar{x}_i} dx dy] \dot{\bar{x}}_i
$$

$$
+ \int \int \sigma \nabla \cdot p\mathbf{S} \, dx dy = 0 \tag{3.3}
$$





Here, the integral range is kept implicit, and weight $\sigma$ is given without a subscript for simplicity. By noting the relations:

$$\frac{\partial p}{\partial \lambda_1} = -(x - \bar{x})^2 p,$$

$$\frac{\partial p}{\partial \lambda_2} = -(y - \bar{y})^2 p,$$

$$\frac{\partial p}{\partial \lambda_3} = -(x - \bar{x})(y - \bar{y})p,$$

$$\frac{\partial p}{\partial \bar{x}} = [2\lambda_1(x - \bar{x}) + \lambda_3(y - \bar{y})]p,$$

$$\frac{\partial p}{\partial \bar{y}} = [2\lambda_2(y - \bar{y}) + \lambda_3(x - \bar{x})]p,$$

Eq. (3.3) further reduces to:

$$\dot{\lambda}_1[-\langle(x-\bar{x})^2\sigma\rangle + \langle\sigma\rangle\langle(x-\bar{x})^2\rangle] + \dot{\lambda}_2[-\langle(y-\bar{y})^2\sigma\rangle + \langle\sigma\rangle\langle(y-\bar{y})^2\rangle] + \dot{\lambda}_3[-\langle(x-\bar{x})(y-\bar{y})\sigma\rangle + \langle\sigma\rangle\langle(x-\bar{x})(y-\bar{y})\rangle]$$

$$+\dot{\bar{x}}[2\lambda_1\langle(x-\bar{x})\sigma\rangle + \lambda_3\langle(y-\bar{y})\sigma\rangle] + \dot{\bar{y}}[2\lambda_2\langle(y-\bar{y})\sigma\rangle + \lambda_3\langle(x-\bar{x})\sigma\rangle] = \langle\mathbf{S}\cdot\nabla\sigma\rangle, \quad (3.4)$$

where $\langle\ \rangle$ suggests the phase–space average.

For proceeding further, we note that the moments are given by

$$\langle x - \bar{x}\rangle = \langle y - \bar{y}\rangle = 0 \tag{3.5a}$$

$$\langle(x-\bar{x})^2\rangle = \frac{1}{2\lambda_1}(1 - \frac{\lambda_3^2}{4\lambda_1\lambda_2})^{-1} \tag{3.5b}$$

$$\langle(y-\bar{y})^2\rangle = \frac{1}{2\lambda_2}(1 - \frac{\lambda_3^2}{4\lambda_1\lambda_2})^{-1} \tag{3.5c}$$

$$\langle(x-\bar{x})(y-\bar{y})\rangle = -\frac{\lambda_3}{4\lambda_1\lambda_2}(1 - \frac{\lambda_3^2}{4\lambda_1\lambda_2})^{-1} \tag{3.5d}$$

$$\langle(x-\bar{x})^3\rangle = \langle(y-\bar{y})^3\rangle = \langle(x-\bar{x})^2(y-\bar{y})\rangle = \langle(x-\bar{x})(y-\bar{y})^2\rangle = 0 \tag{3.5e}$$

$$\langle(x-\bar{x})^4\rangle = \frac{3}{4\lambda_1^2}(1 - \frac{\kappa}{4})^{-2} \tag{3.5f}$$

$$\langle(y-\bar{y})^4\rangle = \frac{3}{4\lambda_2^2}(1 - \frac{\kappa}{4})^{-2} \tag{3.5g}$$

$$\langle(x-\bar{x})^2(y-\bar{y})^2\rangle = \frac{1}{4\lambda_1\lambda_2}(1 + \frac{\kappa}{2})(1 - \frac{\kappa}{4})^{-2} \tag{3.5h}$$

$$\langle(x-\bar{x})^3(y-\bar{y})\rangle = -\frac{3\lambda_3}{8\lambda_1^2\lambda_2}(1 - \frac{\kappa}{4})^{-2} \tag{3.5i}$$

$$\langle(x-\bar{x})(y-\bar{y})^3\rangle = -\frac{3\lambda_3}{8\lambda_1\lambda_2^2}(1 - \frac{\kappa}{4})^{-2} \tag{3.5j}$$

as derived in the Appendix A.2, where

$$\kappa = \frac{\lambda_3^2}{\lambda_1\lambda_2} \tag{3.5k}$$

and the condition $4 > \kappa$ is required to ensure positive variances. To obtain the results for the third moments, we also need to assume $\lambda_1\lambda_2(4\lambda_1\lambda_2 - 3\lambda_3^2)^2 - 6\lambda_3^6 \neq 0$. See the Appendix for the derivations.





By using these expressions (3.5a–j) for the moments, we apply varying weights, $\sigma$, on Eq. (3.4). By setting $\sigma = x - \bar{x}$ and $\sigma = y - \bar{y}$, respectively, we obtain

$$\dot{\bar{x}} = \langle S_x \rangle, \tag{3.6a}$$

$$\dot{\bar{y}} = \langle S_y \rangle. \tag{3.6b}$$

The physical meaning of the results are clear: we recover the mean equations for the evolution.

Next, we set $\sigma = (x - \bar{x})^2$, $\sigma = (y - \bar{y})^2$, and $\sigma = (x - \bar{x})(y - \bar{y})$ in Eq. (3.4). These leads to:

$$-\frac{\dot{\lambda}_1}{\lambda_1} - \frac{\kappa}{4}\frac{\dot{\lambda}_2}{\lambda_2} + \frac{\kappa}{2}\frac{\dot{\lambda}_3}{\lambda_3} = 4\lambda_1(1 - \frac{\kappa}{4})^2 \langle (x - \bar{x})S_x \rangle \tag{3.7a}$$

$$-\frac{\kappa}{4}\frac{\dot{\lambda}_1}{\lambda_1} - \frac{\dot{\lambda}_2}{\lambda_2} + \frac{\kappa}{2}\frac{\dot{\lambda}_3}{\lambda_3} = 4\lambda_2(1 - \frac{\kappa}{4})^2 \langle (y - \bar{y})S_y \rangle \tag{3.7b}$$

$$\frac{\dot{\lambda}_1}{\lambda_1} + \frac{\dot{\lambda}_2}{\lambda_2} - (1 + \frac{\kappa}{4})\frac{\dot{\lambda}_3}{\lambda_3} = \frac{4\lambda_3}{\kappa}(1 - \frac{\kappa}{4})^2 \langle (y - \bar{y})S_x + (x - \bar{x})S_y \rangle \tag{3.7c}$$

where $\kappa$ has been defined by Eq. (3.5k). Eqs. (3.7a, b, c) can be re-written into three separate prognostic equations for $\lambda_j$ ($j = 1, 2, 3$) by a matrix inversion of the left hand side.

To see this last procedure, we may re–write Eqs. (3.7a, b, c) as

$$\alpha_{11}\frac{\dot{\lambda}_1}{\lambda_1} + \alpha_{12}\frac{\dot{\lambda}_2}{\lambda_2} + \alpha_{13}\frac{\dot{\lambda}_3}{\lambda_3} = f_1,$$

$$\alpha_{21}\frac{\dot{\lambda}_1}{\lambda_1} + \alpha_{22}\frac{\dot{\lambda}_2}{\lambda_2} + \alpha_{23}\frac{\dot{\lambda}_3}{\lambda_3} = f_2,$$

$$\alpha_{31}\frac{\dot{\lambda}_1}{\lambda_1} + \alpha_{32}\frac{\dot{\lambda}_2}{\lambda_2} + \alpha_{33}\frac{\dot{\lambda}_3}{\lambda_3} = f_3.$$

The matrix inversion of the above leads to:

$$\frac{\dot{\lambda}_1}{\lambda_1} = \frac{1}{D}[(\alpha_{22}\alpha_{33} - \alpha_{23}\alpha_{32})f_1 + (-\alpha_{12}\alpha_{33} + \alpha_{13}\alpha_{32})f_2 + (\alpha_{12}\alpha_{23} - \alpha_{13}\alpha_{22})f_3], \tag{3.8a}$$

$$\frac{\dot{\lambda}_2}{\lambda_2} = \frac{1}{D}[(\alpha_{23}\alpha_{31} - \alpha_{21}\alpha_{33})f_1 + (-\alpha_{13}\alpha_{31} + \alpha_{11}\alpha_{33})f_2 + (\alpha_{13}\alpha_{21} - \alpha_{11}\alpha_{23})f_3], \tag{3.8b}$$

$$\frac{\dot{\lambda}_3}{\lambda_3} = \frac{1}{D}[(\alpha_{21}\alpha_{32} - \alpha_{22}\alpha_{31})f_1 + (-\alpha_{11}\alpha_{32} + \alpha_{12}\alpha_{31})f_2 + (\alpha_{11}\alpha_{22} - \alpha_{12}\alpha_{21})f_3], \tag{3.8c}$$

where

$$D = \alpha_{11}(\alpha_{22}\alpha_{33} - \alpha_{23}\alpha_{32}) + \alpha_{12}(-\alpha_{21}\alpha_{33} + \alpha_{23}\alpha_{31}) + \alpha_{13}(\alpha_{21}\alpha_{32} - \alpha_{22}\alpha_{31})$$

$$= \alpha_{21}(\alpha_{32}\alpha_{13} - \alpha_{33}\alpha_{12}) + \alpha_{22}(-\alpha_{31}\alpha_{13} + \alpha_{33}\alpha_{11}) + \alpha_{23}(\alpha_{31}\alpha_{12} - \alpha_{32}\alpha_{11})$$

$$= \alpha_{31}(\alpha_{12}\alpha_{23} - \alpha_{13}\alpha_{22}) + \alpha_{32}(-\alpha_{11}\alpha_{23} + \alpha_{13}\alpha_{21}) + \alpha_{33}(\alpha_{11}\alpha_{22} - \alpha_{12}\alpha_{21}). \tag{3.8d}$$





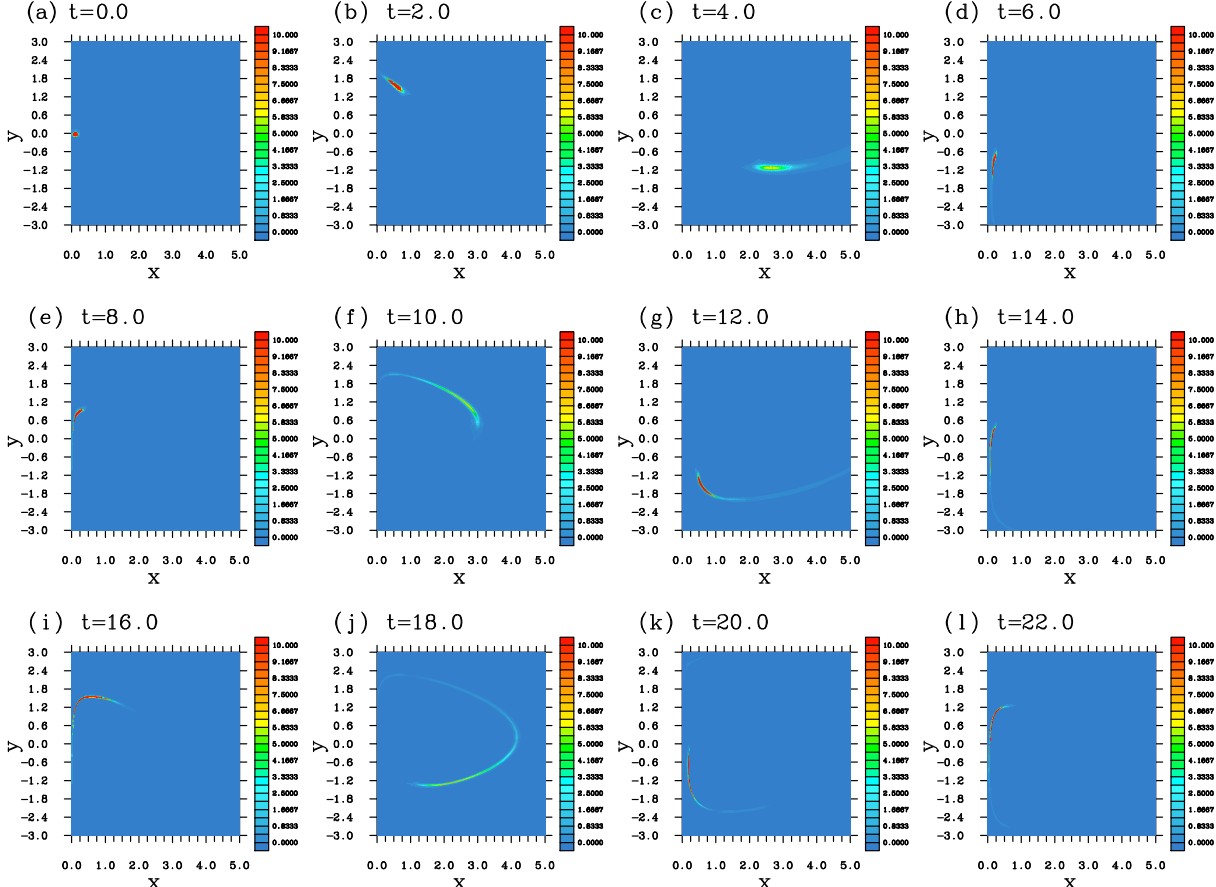

**Figure 1.** Snap shots of evolution of PDF with the convective energy cycle by a direct numerical integration of the Liouville equation.

In the following two sections, we apply those general formulations developed over these last two sections to the two specific systems: (i) the convective energy–cycle system introduced by Yano and Plant (2012) in Sec. 4, and (ii) the Lorenz's (1963) strange–attractor system in Sec. 5. The derived prognostic equations for the PDF parameters are integrated by the 4th–order Runge–Kutta method with a time step depending the assumed–PDF model.

## 4 Convective Energy-Cycle System (Yano and Plant 2012)

The convective energy–cycle system introduced by Yano and Plant (2012) is presented in a nondimensinoal form by:

$$\dot{x} = xy, \tag{4.1a}$$

$$\dot{y} = -x + 1, \tag{4.1b}$$



where $x$ is the convective kinetic energy (mass flux), and $y$ the cloud work function. The equilibrium is at $(x, y) = (1, 0)$, and $x > 0$ always.

An explicit calculation of the evolution of the initial uncertainty distribution with this system by the Liouville equation has
230 already been peformed by Yano and Ouchter (2017). Their result is reproduced in a different format than in the original paper in Fig. 1. Here, the initial condition is a very localized Gaussian distribution as shown in Fig. 1(a). Note that the characteristics of the subsequent evolution are qualitatively different from those of the assumed PDF forms introduced in the following. The main challenge of the assumed–PDF approach is, nevertheless, to predict the bulk statistics of the system fairly accurately.

### 4.1   Combination of the Gamma and the Gauss distributions (Model I)

Sine the system is semi–infinite in the $x$ direction, whereas $y$ extends to infinity to both sides, thus by following the argument in YLP, the most natural choice for the assumed form of PDF for this system is to adopt the gamma distribution in the $x$ direction and the Gaussian in $y$ direction, which is referred as the model I in the following. Thus,

$$p(x, y) = p_1(x) p_2(y), \tag{4.1.1}$$

where

$$p_1(x) = p_{10} x^\mu \exp(-\lambda_1 x), \quad p_{10} = \frac{\lambda_1^{\mu+1}}{\Gamma(\mu+1)}, \tag{4.1.2a}$$

$$p_2(y) = p_{20} \exp[-\lambda_2 (y - \bar{y})^2], \quad p_{20} = (\lambda_2/\pi)^{1/2} \tag{4.1.2b}$$

($cf.$, Appendix A.1). In general, the Liouville equation leads to Eq. (2.1.6), which in this case, reduces to:

$$\dot{\mu}[\int\int \sigma \frac{\partial p}{\partial \mu} dx dy - \int\int \sigma p dx dy \int\int \frac{\partial p}{\partial \mu} dx dy] + \dot{\bar{y}}[\int\int \sigma \frac{\partial p}{\partial \bar{y}} dx dy - \int\int \sigma p dx dy \int\int \frac{\partial p}{\partial \bar{y}} dx dy]$$

$$+ \sum_{i=1}^{2} \dot{\lambda}_i [\int\int \sigma \frac{\partial p}{\partial \lambda_i} dx dy - \int\int \sigma p dx dy \int\int \frac{\partial p}{\partial \lambda_i} dx dy] = \int\int p \mathbf{S} \cdot \nabla \sigma dx dy, \tag{4.1.3}$$

where

$$\frac{\partial p}{\partial \mu} = p \log x, \tag{4.1.4a}$$

$$\frac{\partial p}{\partial \bar{y}} = 2\lambda_2 (y - \bar{y}) p, \tag{4.1.4b}$$

$$\frac{\partial p}{\partial \lambda_1} = -px, \tag{4.1.4c}$$

$$\frac{\partial p}{\partial \lambda_2} = -(y - \bar{y})^2 p \tag{4.1.4d}$$

By substituting Eqs. (4.1.4a, b, c, d) into Eq. (4.1.3), we obtain:

$$\dot{\mu}[\langle \sigma \log x \rangle - \langle \sigma \rangle \langle \log x \rangle] + 2\lambda_2 \dot{\bar{y}}[\langle \sigma(y - \bar{y}) \rangle - \langle \sigma \rangle \langle y - \bar{y} \rangle]$$

$$- \dot{\lambda}_1 [\langle \sigma x \rangle - \langle \sigma \rangle \langle x \rangle] - \dot{\lambda}_2 [\langle \sigma(y - \bar{y})^2 \rangle - \langle \sigma \rangle \langle (y - \bar{y})^2 \rangle] = \langle \mathbf{S} \cdot \nabla \sigma \rangle \tag{4.1.5}$$





Recall that $\langle\ \rangle$ suggests the phase–space average.

We are going to derive prognostic equations for the four parameters, $\mu$, $\bar{y}$, $\lambda_i$ $(i = 1, 2)$ from Eq. (4.1.5) by choosing four

different weights, $\sigma$. Here, the first two only depend on $x$, and the last two only on $y$. The first two choices are $\sigma = x$ and $\sigma = x^2$. Substitutions of these weights into Eq. (4.1.5) lead to:

$$\frac{\dot{\mu}}{\lambda_1} - (\mu + 1)\frac{\dot{\lambda}_1}{\lambda_1^2} = \langle S_x \rangle \tag{4.1.6a}$$

$$\frac{2\mu + 3}{\lambda_1^2}\dot{\mu} - \frac{2(\mu + 2)(\mu + 1)}{\lambda_1^3}\dot{\lambda}_1 = 2\langle xS_x \rangle \tag{4.1.6b}$$

noting $\langle \sigma(y - \bar{y})\rangle - \langle\sigma\rangle\langle y - \bar{y}\rangle = 0$ and $\langle\sigma(y - \bar{y})^2\rangle - \langle\sigma\rangle\langle(y - \bar{y})^2\rangle = 0$ for both $\sigma = x$ and $\sigma = x^2$. We have also noted the

relations:

$$\langle x \log x \rangle = \frac{1}{\lambda_1}[(\mu + 1)\langle \log x \rangle + 1],$$

$$\langle x^2 \log x \rangle = \frac{1}{\lambda_1^2}[(\mu + 1)(\mu + 2)\langle \log x \rangle + 2\mu + 3].$$

By combining Eqs. (4.1.6a, b), we further obtain stand-alone prognostic equations for these two individual parameters:

$$\dot{\mu} = 2[(\mu + 2)\langle S_x \rangle - \lambda_1\langle xS_x \rangle], \tag{4.1.6c}$$

$$\dot{\lambda}_1 = \frac{\lambda_1^2}{\mu + 1}[(2\mu + 3)\langle S_x \rangle - 2\lambda_1\langle xS_x \rangle]. \tag{4.1.6d}$$

Here,

$$S_x = xy \tag{4.1.7a}$$

$$\langle S_x \rangle = \langle xy \rangle = \langle x \rangle\langle y \rangle = \frac{\mu + 1}{\lambda_1}\bar{y} \tag{4.1.7b}$$

$$\langle xS_x \rangle = \langle x^2y \rangle = \langle x^2 \rangle\langle y \rangle = \frac{(\mu + 2)(\mu + 1)}{\lambda_1^2}\bar{y} \tag{4.1.7c}$$

In deriving these three results (4.1.7a, b, c), we have used the relaions

$$\langle x \rangle = \frac{\mu + 1}{\lambda_1}, \quad \langle x^2 \rangle = \frac{(\mu + 2)(\mu + 1)}{\lambda_1^2}, \quad \langle y \rangle = \bar{y}.$$

Using these three relations, the right hand side of Eq. (4.1.6c) vanishes, and

$$\dot{\mu} = 0, \tag{4.1.8a}$$

wherreas Eq. (4.1.6d) reduces to:

$$\dot{\lambda}_1 = -\lambda_1\bar{y}. \tag{4.1.8b}$$





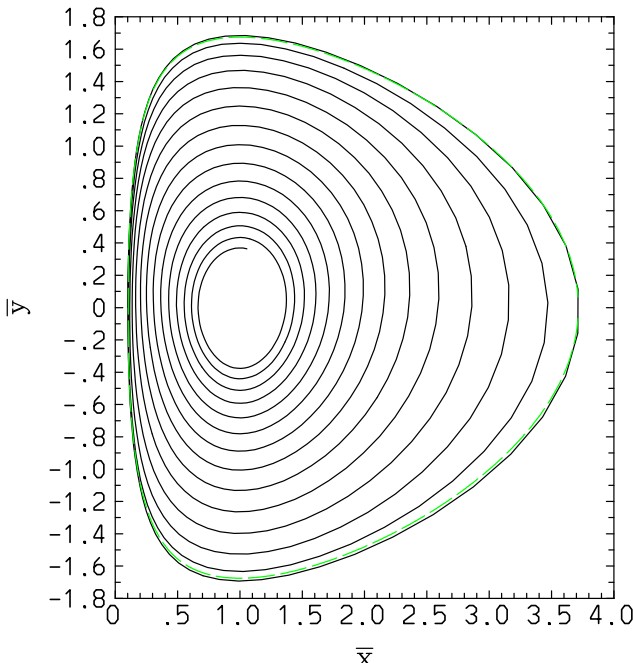

**Figure 2.** Trajectory of $(\bar{x}, \bar{y})$: as directly obtained from the Liouville equation (black solid) and by the assumed PDF (Eqs. 4.1.1, 4.1.2a, b)a model I (green dash) results. Here, $\mu = 10.2271805$ and $R = 2.19206810$ in Eq. (4.1.9). This assumed PDF solution simply takes a closed orbit, failing to re–produce a damping tendency of the actual PDF evolution.

As for the other two weights depending only on $y$, we choose $\sigma = y - \bar{y}$ and $\sigma = (y - \bar{y})^2$. In reductions, we invoke the relations

$$\langle y - \bar{y} \rangle = \langle (y - \bar{y})^3 \rangle = 0, \quad \langle (y - \bar{y})^2 \rangle = \frac{1}{2\lambda_2}, \quad \langle (y - \bar{y})^4 \rangle = \frac{3}{4\lambda_2^2}.$$

The final results are:

$$\dot{\bar{y}} = 1 - \frac{\mu + 1}{\lambda_1}, \tag{4.1.8c}$$

$$\dot{\lambda}_2 = 0. \tag{4.1.8d}$$

Thus, Eqs. (4.1.8a, b, c, d) dictate the evolution of the distribution (4.1.1), in which the two parameters, $\mu$ and $\lambda_2$, turn out to be constants of time. On the other hand, by combining Eqs. (4.1.8b) and (4.1.8c), we find

$$-\frac{d\lambda_1}{\lambda_1 \bar{y}} = \frac{d\bar{y}}{1 - \frac{\mu + 1}{\lambda_1}} = dt$$

The first two terms can be re-arranged to:

$$-\frac{1}{\lambda_1}(1 - \frac{\mu + 1}{\lambda_1})d\lambda_1 = \bar{y}d\bar{y},$$





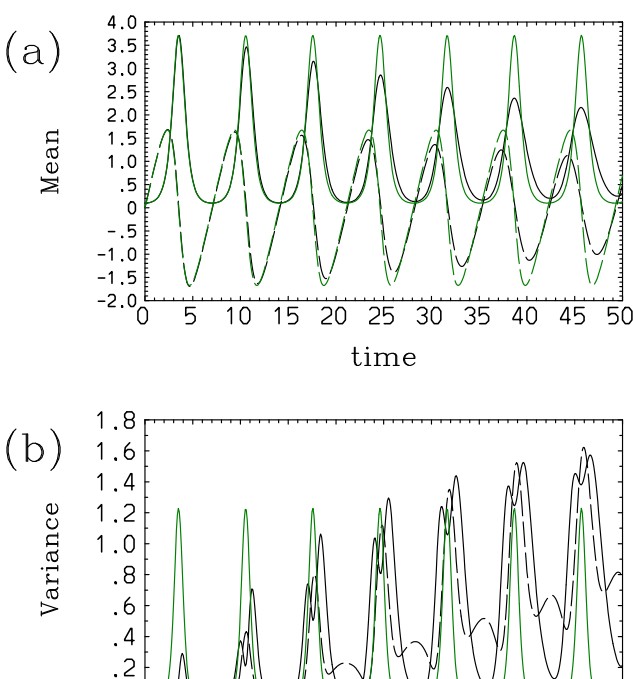

**Figure 3.** Comparisons of the statistics of the convective energy cycle between results with a direct computation of the Liouville equation (black) and that based on the assumed–PDF method (model I: green): (a) Means, $\bar{x}$ (solid) and $\bar{y}$ (long dash), (b) Variances, $\langle (x-\bar{x})^2 \rangle$ (solid) and $\langle (y-\bar{y})^2 \rangle$ (long dash). :To Be Done: Captilize titley

which can readily be integrated, and the trajectory $(\bar{x}, \bar{y})$ of the system is, noting also that $\bar{x} = (\mu + 1)/\lambda_1$, found as:

$$2\bar{x} - 2\log \bar{x} + \bar{y}^2 = R \tag{4.1.9}$$

where $R$ is a constant, noting that $\mu$ is constant with time by Eq. (4.1.8a). By comparing this final expression with Eq. (25) of Yano and Plant (2012), we find that the mean trajectory is identical to that of the solution of the system (4.1a, b). Note that this is not necessarily the case. In fact, the numerical result in Fig. 2 shows that a full solution presents a damping circular trajectory in the phase space of $(\bar{x}, \bar{y})$ towards the equilibrium $(1, 0)$. This aspect is simply not captured by the given assumed PDF.

Fig. 3 shows further statistical quantifications on the performance of the model I. Here, the model time step is $\Delta t = 1 \times 10^{-2}$. As already remarked, this model predicts a simple periodic cycle for the mean values, and fails to re–produce a damping tendency of the actual PDF evolution (Fig. 3(a)). The same follows with $x$–variance (Fig. 3(b)), whereas the model I does not predict the evolution of the $y$–variance, being constant with time by Eq. (4.1.8d). An obvious defect of this assumed form is traced to the lack of correlation between the two variables, and the key nonlinear contribution, $\langle x'y' \rangle$, drops out from the set of

evolution equations, where $x' = x - \langle x \rangle$ and $y' = y - \langle y \rangle$.





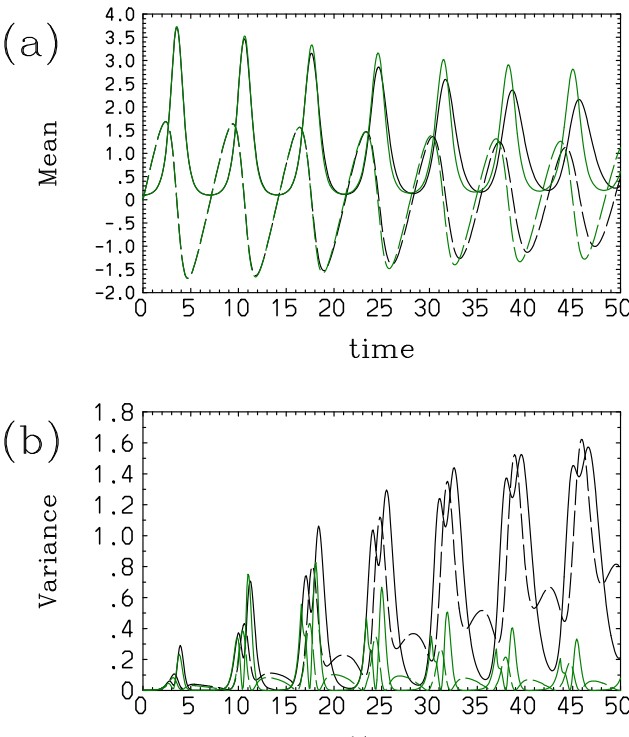

**Figure 4.** Comparisons of the statistics of the convective energy cycle between results with a direct computation of the Liouville equation (black) and that based on the assumed–PDF method (model II: green): (a) Mean, $\bar{x}$ (solid) and $\bar{y}$ (long dash), (b) Variance, $\langle (x - \bar{x})^2 \rangle$ (solid) and $\langle (y - \bar{y})^2 \rangle$ (long dash). :To Be Done: Captilize titley

### 4.2 Two–Dimensional Gaussian Distribution (Model II)

The model I (Eq. 4.1.1) in the last section fails to predict a dissipating tendency of mean and amplifying tendency of the variance. This reason is traced to lack of correlation between the two dependent variables, $x$ and $y$, in distribution, thus the problem of the prediction of the mean becomes identical of that of the original dynamical system.

The most straightforward modification to overcome this issue is to modify the distribution into:

$$p(x,y) = p_0 x^\mu \exp[-\lambda_1 x - \lambda_2 (y - \bar{y})^2 - \lambda_3 x (y - \bar{y})].$$

However, this assumed PDF has an unfavorable feature that the resulting integrals become not possible to be performed analytically any more. Need for numerical integrals add up a computation cost, and effectively kills an advantage of the assumed PDF approach.

To avoid this difficulty, we instead adopt a Gaussian distribution with two variables considered in Sec. II.1 (Eq. 3.1): this model is referred as the model II. A minor disadvantage with this assumed PDF is that a distribution can spread to $x < 0$.





However, this disadvantage has no serious consequence so long as we focus our attentions to the basic statistics, mean and variance (and also correlation), and the mean of $x$ remains positive, $i.e.,$ $\bar{x} > 0$.

By applying the expression of the source term for this system (4.1a, b) to general formulas in Sec. II.1, we obtain the

following prognostic equation set for the assumed–PDF parameters:

$$\dot{\bar{x}} = \bar{x}\bar{y} - \frac{\kappa}{4\lambda_3}(1 - \frac{\kappa}{4})^{-1} \tag{4.2.1a}$$

$$\dot{\bar{y}} = -\bar{x} + 1 \tag{4.2.1b}$$

$$-\frac{\dot{\lambda}_1}{\lambda_1} - \frac{\kappa}{4}\frac{\dot{\lambda}_2}{\lambda_2} + \frac{\kappa}{2}\frac{\dot{\lambda}_3}{\lambda_3} = (1 - \frac{\kappa}{4})(-\frac{\lambda_3}{\lambda_2}\bar{x} + 2\bar{y}) \tag{4.2.1c}$$

$$-\frac{\kappa}{4}\frac{\dot{\lambda}_1}{\lambda_1} - \frac{\dot{\lambda}_2}{\lambda_2} + \frac{\kappa}{2}\frac{\dot{\lambda}_3}{\lambda_3} = (1 - \frac{\kappa}{4})\frac{\lambda_3}{\lambda_1} \tag{4.2.1d}$$

$$\frac{\dot{\lambda}_1}{\lambda_1} + \frac{\dot{\lambda}_2}{\lambda_2} - (1 + \frac{\kappa}{4})\frac{\dot{\lambda}_3}{\lambda_3} = (1 - \frac{\kappa}{4})(-\frac{2\lambda_2}{\lambda_3} + \frac{2\lambda_1}{\lambda_3}\bar{x} - \bar{y}) \tag{4.2.1e}$$

Recall that $\kappa$ is defined by Eq. (3.5k):

$$\kappa = \frac{\lambda_3^2}{\lambda_1\lambda_2} \tag{4.2.2}$$

The initial condition in this case is set equal to that of the full Liouville run, which is also initialized with a Gaussian.

Characteristics of the evolution of the system with this model II obtained with the time step of $\Delta t = 1 \times 10^{-4}$ are shown in

Fig. 4: the mean values (a) damp as the case with the explicit Liouville run. The agreements are almost perfect up to the end of the third cycle, but then a difference gradually becomes noticeable. A periodic increase of variances (b) are also predicted up to the second cycle. However, after the end of the third cycle, the variances predicted by the model II rapidly decrease with time. The model is numerically unstable, and blows up after $t = 60$ with $\Delta t = 10^{-2}$. Euler time stepping is also attempted: it crushes at $t = 23.5$. Those behaviors can be understood, to a good extent, simply by inspecting the obtained prediction equations. Even

under the model II, the evolution of $\bar{y}$ (Eq. 4.2.1b) still follows that of $y$ replacing the terms by the averages as also the case with the model I. However, an extra term is found for the evolution equation for $\bar{x}$ (Eq. 4.2.1a), which can work as a damping term whenever $1 - \kappa/4 > 0$ is satisfied, as expected from the full Liouville simulation. Due to this dissipating tendency of $\bar{x}$, $\bar{y}$ also dissipates with time to the extent that the former dissipates, as seen in the long–dash curves in Fig. 4(a).

The condition with $1 - \kappa/4 > 0$ is indeed satisfied initially with a Gaussian distribution leading to $\kappa = 0$, with $\lambda_3 = 0$.

However, the distribution evolves associated with the increasing $\kappa$ with time, approaching towards $\kappa = 4$. Thus, this dissipative term also becomes smaller with time, and at a certain point, $\bar{x}$ no longer dissipates as effectively as seen in the full Liouville solution, as seen in the solid curves in Fig. 4(a). The variances also grow with time so long as $1 - \kappa/4 > 0$ by following Eqs. (4.2.1c, d, e). However, as remarked, due to the tendency of $\kappa \to 4$, this growing tendency only continues over the first three convective cycles, and then the variances begin to diminish with time, as seen in Fig. 4(b). These result suggests that

adding a cross term ($i.e.,$ $\lambda_3 \neq 0$) in the distribution is not enough to reproduce statistical tendencies for an extensive duration of the simulation, especially because the variances tend to collapse after initial realistic tendencies of the growths.



### 4.3 Alternative possibility

As an alternative possibility of distributions, the form,

$$
p(x,y) = \begin{cases} p_0 x^\mu \exp(-\lambda_1 x - \lambda_2^+ y) & y > 0 \\ p_0 x^\mu \exp(-\lambda_1 x + \lambda_2^- y) & y < 0, \end{cases}
$$

is also considered. As it turns out, evolution of the mean values, $\bar{x}$ and $\bar{y}$, remains identical to that of the model I without any damping tendency. A slight improvement is that the standard deviation in $y$, in this case, evolves with:

$$
\frac{d}{dt}\langle y'^2 \rangle = \langle y \rangle \langle S_y \rangle
$$

Due to these only limited improvements expected, this case is not actually attempted. It transpires that it is crucial to include a dependence on $xy$ in the distribution for successfully predicting a damping tendency of mean values as the case with the model II.

### 5 Lorenz (1963) System

The system proposed by Lorenz (1963) is given by:

$$
\dot{x} = -Px + Py, \tag{5.1a}
$$

$$
\dot{y} = -xz + rx - y, \tag{5.1b}
$$

$$
\dot{z} = xy - bz, \tag{5.1c}
$$

in which assume the standard parameters: $b = 8/3$, the Rayleigh number, $r = 28$, and Prandtl number, $P = 10$. The system consists of three unstable steady solutions (*i.e.*, fixed points): two of them corresponding to steady convection are found at $(x,y,z) = (\pm 6\sqrt{2}, \pm 6\sqrt{2}, 27)$, and another is found at $(x,y,z) = (0,0,0)$.

First, as a reference, the evolution of PDF is computed by directly integrating the Liouville equation. Here, the adopted numerics are identical to those adopted in Yano and Phillips (2016) for the Fokker–Planck equation, except for a treatment for the diffusion term raises from stochasticity is missing in the present case. The initial condition is a Gaussian distribution centered at $(0,1,0)$ with the variance, 12.5, in all three directions. The initial center point is the identical to the initial condition adopted by Lorenz (1963). An identical run has also been repeated by replacing the initial center of the Gaussian distribution of the origin, $(0,0,0)$, of the system. Some remarks will be also added in the following on this latter case.

Figs. 5 and 6 show that evolution of PDF is highly non–Gaussian both on $y$–$z$ and $x$–$y$ planes, respectively. On both planes, the distribution splits into two peaks over time upto $t = 1$, corresponding to two unstable fixed points of the system. Distributions around these two peaks gradually diffuse with time as the stranger attractor fully develops. Note that the Lorenz system is chaotic, thus the individual trajectories of solutions remain nonstationary throughout their evolutions. However, as expected from the evolution of the distribution, the statistics of the system converge to a stationary state after an initial transient



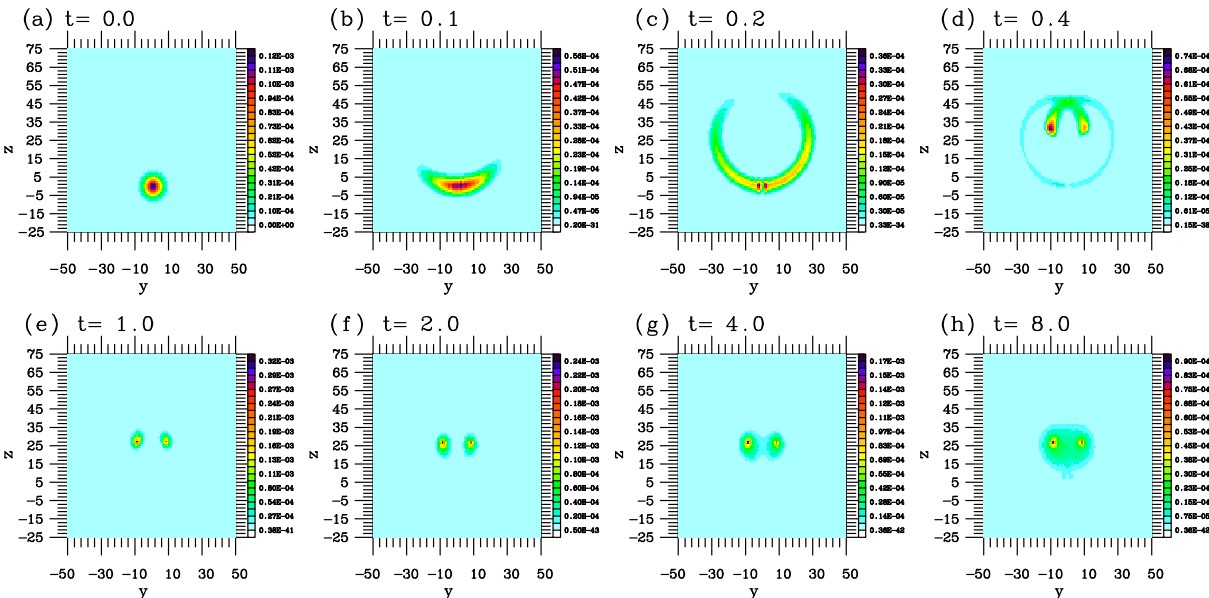

**Figure 5.** Time evolution of PDF for the Lorenz system on $y$–$z$ plane as directly predicted by the Liouville equation

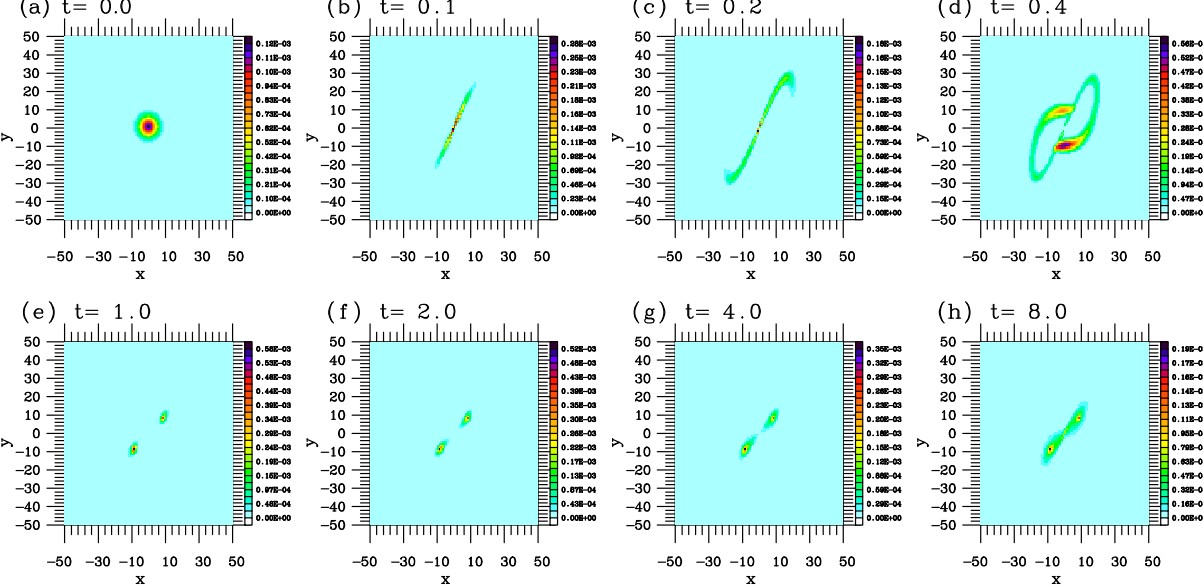

**Figure 6.** The same as Fig. 5 but on $x$–$y$ plane.

period gradually. Note that this is a major difference from a fully turbulent system to a low–dimensional chaotic system: in the former case, the statistics can also remain nonstationary throughout the evolution of the system. As going to be demonstrated in the following, this aspect turns out to be the hardest to reproduce by an assumed–PDF form only with a few parameters,





because a governing equation system predicting those PDF parameters itself can become a nonlinear chaotic system. This tendency most clearly emerges with the model I in the following. In the following assumed–PDF demonstrations, the time step
is by default $\Delta t = 10^{-4}$ is adopted. Runs have also been repeated with $\Delta t = 2 \times 10^{-4}$ to find no modification in results.

## 5.1   Lorenz (1963) System: Gaussian Distribution (Model I)

As a first model (model I), we consider a three–dimensional Gaussian distribution, but without correlations between the dependent variables, $x$, $y$, and $z$:

$$p(x,y,z) = p_1(x)p_2(y)p_3(z), \tag{5.1.1}$$

where

$$p_1(x) = p_{10}\exp[-\lambda_1(x-\bar{x})^2], \tag{5.1.2a}$$
$$p_2(y) = p_{20}\exp[-\lambda_2(y-\bar{y})^2], \tag{5.1.2b}$$
$$p_3(z) = p_{30}\exp[-\lambda_3(z-\bar{z})^2], \tag{5.1.2c}$$

in which normalization conditions are:

$$p_{j0} = (\lambda_j/\pi)^{1/2} \tag{5.1.3}$$

with $j = 1, 2, 3$.

From the general formulation (2.1.6), we obtain:

$$\sum_{j=1}^{3}\{\dot{\lambda}_j[\int \sigma\frac{\partial p}{\partial \lambda_j}d\mathbf{x} - \int \sigma p d\mathbf{x}\int \frac{\partial p}{\partial \lambda_j}d\mathbf{x}] + \dot{\bar{x}}_j[\int \sigma\frac{\partial p}{\partial \bar{x}_j}d\mathbf{x} - \int \sigma p d\mathbf{x}\int \frac{\partial p}{\partial \bar{x}_j}d\mathbf{x}]\} = \int p\mathbf{S}\cdot\frac{\partial \sigma}{\partial \mathbf{x}}d\mathbf{x}, \tag{5.1.4}$$

whrere $\mathbf{x} = (x_1, x_2, x_3) = (x, y, z)$, $\sigma$ is a weight, and we find the relations:

$$\partial p/\partial \lambda_j = -(x_j - \bar{x}_j)^2 p, \tag{5.1.5a}$$
$$\partial p/\partial \bar{x}_j = 2\lambda_j(x_j - \bar{x}_j)p. \tag{5.1.5b}$$

By substituting them into Eq. (5.1.4):

$$\sum_{j=1}^{3}\{\dot{\lambda}_j[-\langle\sigma(x_j-\bar{x}_j)^2\rangle + \langle\sigma\rangle\langle(x_j-\bar{x}_j)^2\rangle] + 2\lambda_j\dot{\bar{x}}_j\langle\sigma(x_j-\bar{x}_j)\rangle\} = \langle\mathbf{S}\cdot\frac{\partial\sigma}{\partial\mathbf{x}}\rangle. \tag{5.1.6}$$

We choose the weights as $\sigma = x_j - \bar{x}_j$ and $\sigma = (x_j - \bar{x}_j)^2$ ($j = 1, 2, 3$), then (5.1.6) recudes to:

$$\dot{\bar{x}}_j = \langle S_j\rangle, \tag{5.1.7a}$$
$$\frac{d}{dt}\left(\frac{1}{\lambda_j}\right) = 4\langle(x_j - \bar{x}_j)S_j\rangle \tag{5.1.7b}$$





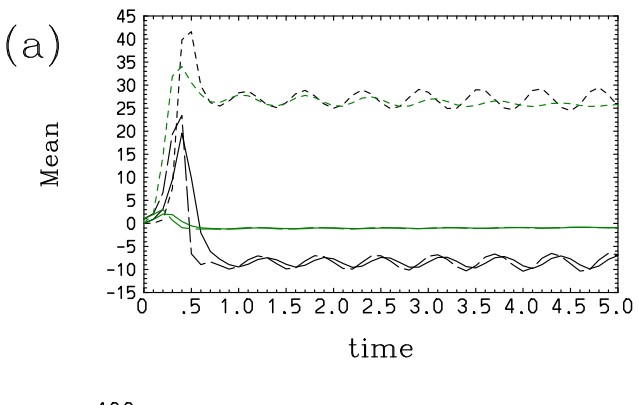

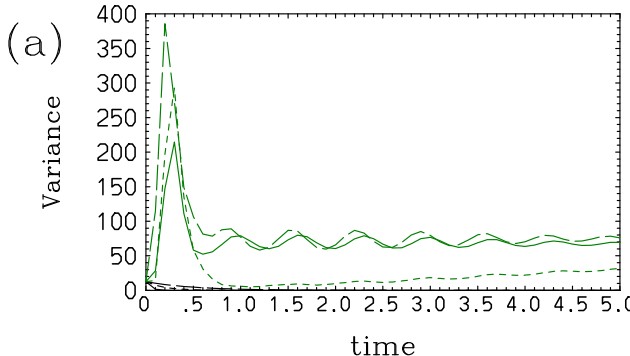

**Figure 7.** Statistics of the Lorenz system with the model I: model results (black), and those by direct prediction with the Liouville equation (green):(a) mean and (b) the variance. Here, and in the following plots, the curves are defined for the $x-$, $y-$, and $z-$components are in solid, long dash, and short dash.

for $j = 1, 2, 3$. From Eq. (5.1a, b, c), the source terms are defined by

$$S_x = -Px + Py,$$
$$S_y = -xz + rx - y,$$
$$S_z = xy - bz.$$

Here,

$$\langle S_j \rangle = S_j(\bar{\mathbf{x}}), \tag{5.1.8}$$

and we also find by direct manipulations:

$$\langle (x - \bar{x})S_x \rangle = -P/2\lambda_1, \tag{5.1.9a}$$
$$\langle (y - \bar{y})S_y \rangle = -1/2\lambda_2, \tag{5.1.9b}$$
$$\langle (z - \bar{z})S_z \rangle = -b/2\lambda_3. \tag{5.1.9c}$$





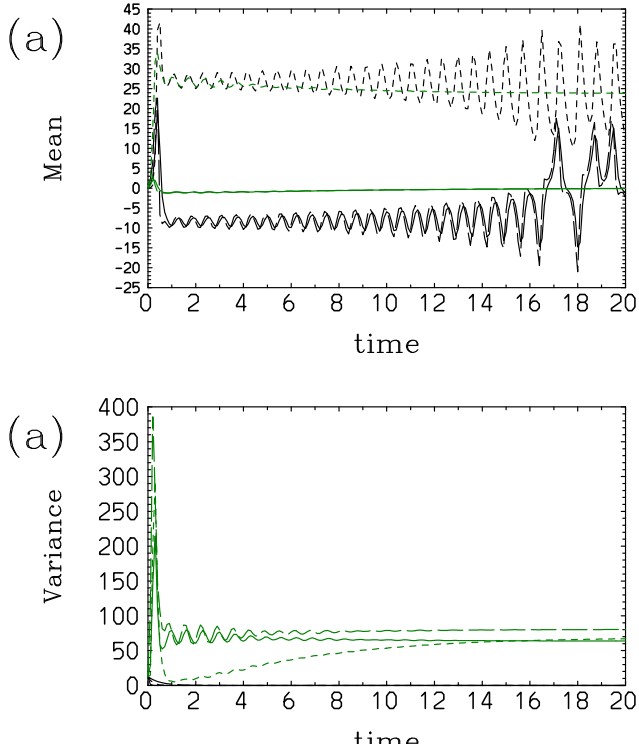

**Figure 8.** The same as Fig. 7 but for a longer duration.

The model I predicts the evolution of $\bar{z}$ (short dash) reasonably, but $\bar{x}$ (solid) and $\bar{y}$ (long dash), somehow, settle to one of two unstable fixed points (to the negative side: in black), without suggesting an alternative possibility. Equal probability among these two fixed points makes the mean values close to zero in the explicit simulation (green: Fig. 7(a)). The model I also fails

to predict an increase of the variances at an initial phase seen in the explicit simulation (green), but they simply decay rapidly (black: Fig. 7(b)). The latter behavior with the variances is already seen in Eq. (5.1.7b), along with the definitions (5.1.9a, b, c) of the right–hand side forcing term: since the parameters, $\lambda_j$ ($j = 1, 2, 3$), are defined to be positive definite, the variance can only decay with time.

Over the longer time, the aforementioned chaotic nature of the system emerges, as shown in Fig. 8. Note especially that the

415 equation set for the means is identical to the original Lorenz's attractor system, thus the means never converge to a statistical equilibrium, as realized in a direct computation of the Liouville equation. Instead, after $t = 5$, the means gradually begin to oscillate around the tentatively–settled unstable fixed points. A first transition happens a little after $t = 16$, over which $\bar{x}$ and $\bar{y}$ transit to an oscillation around the origin.





## 5.2 Lorenz (1963) System: Semi–Exponential in $y$–Direction (Model II)

The attempt of the last subsection, constraining the system in terms of the domain–averaged statistics, has failed to capture a tendency of the system to settle around two unstable fixed points, but the assumed PDF tends to settle only to one of those two fixed points. As a measure of alleviating this tendency, in this subsection, we constrain the PDF in terms of the averages over subdomains of the system. As the model II, we now more specifically constraint the system by:

$$\bar{y}^+ = \langle y \rangle_+ \equiv \int\limits_{0}^{+\infty} y p_2 \, dy \Big/ \int\limits_{0}^{+\infty} p_2 \, dy,$$

$$\bar{y}^- = \langle y \rangle_- \equiv \int\limits_{-\infty}^{0} y p_2 \, dy \Big/ \int\limits_{-\infty}^{0} p_2 \, dy.$$

These constraints suggest the semi–exponential distribution form in $y$–direction:

$$p_2 = \begin{cases} p_{20}^+ \exp(-\lambda_2^+ y), & y > 0 \\ p_{20}^- \exp(\lambda_2^- y), & y < 0 \end{cases} \tag{5.2.1}$$

and at $y = 0$,

$$p_{20}^+ = p_{20}^- = p_{20}. \tag{5.2.2}$$

Here, $p_1$ and $p_3$ in Eq. (5.1.1) remains the same as in model I of Sec. 5.1.

From the normalization condition

$$\int\limits_{-\infty}^{\infty} p_2 \, dy = 1,$$

we find

$$p_{20} = (1/\lambda_2^+ + 1/\lambda_2^-)^{-1}. \tag{5.2.3}$$

Also let:

$$p_+ \equiv \int\limits_{0}^{+\infty} p \, dy = \frac{p_{20}}{\lambda_2^+} \tag{5.2.4a}$$

$$p_- \equiv \int\limits_{-\infty}^{0} p \, dy = \frac{p_{20}}{\lambda_2^-} \tag{5.2.4b}$$

Note further,

$$\frac{\partial p}{\partial \lambda_2^+} = -y p, \ y > 0, \tag{5.2.5a}$$

$$\frac{\partial p}{\partial \lambda_2^-} = y p, \ y < 0, \tag{5.2.5b}$$





and

$$\bar{y}_+ = \int_0^{+\infty} ypdy \int_0^{+\infty} pdy = \frac{1}{\lambda_2^+}, \tag{5.2.6a}$$

$$\bar{y}_- = \int_{-\infty}^0 ypdy \int_{-\infty}^0 pdy = -\frac{1}{\lambda_2^-}, \tag{5.2.6b}$$

$$\bar{y} \equiv \int_{-\infty}^{+\infty} ypdy = \frac{1}{\lambda_2^+} - \frac{1}{\lambda_2^-}. \tag{5.2.6c}$$

The prognostic equations for $\lambda_1$, $\lambda_2^{\pm}$, $\lambda_3$, and $\bar{x}_j$ $(j = 1, \ldots, 3)$ are:

$$\sum_{j=1,3} \Big\{ \dot{\lambda}_j \big[ \int_+ \sigma \frac{\partial p}{\partial \lambda_j} d\mathbf{x} - \int_+ \sigma p d\mathbf{x} \int \frac{\partial p}{\partial \lambda_j} d\mathbf{x} \big] + \dot{\bar{x}}_j \big[ \int_+ \sigma \frac{\partial p}{\partial \bar{x}_j} d\mathbf{x} - \int_+ \sigma p d\mathbf{x} \int \frac{\partial p}{\partial \bar{x}_j} d\mathbf{x} \big] \Big\}$$

$$+ \dot{\lambda}_2^+ \big[ \int_+ \sigma \frac{\partial p}{\partial \lambda_2^+} d\mathbf{x} - \int_+ \sigma p d\mathbf{x} \int_+ \frac{\partial p}{\partial \lambda_2^+} d\mathbf{x} \big] - \dot{\lambda}_2^- \int_+ \sigma p d\mathbf{x} \int_- \frac{\partial p}{\partial \lambda_2^-} d\mathbf{x} + \int_+ \sigma \frac{\partial}{\partial \mathbf{x}} (p\mathbf{S}) d\mathbf{x} = 0, \tag{5.2.7a}$$

$$\sum_{j=1,3} \Big\{ \dot{\lambda}_j \big[ \int_- \sigma \frac{\partial p}{\partial \lambda_j} d\mathbf{x} - \int_- \sigma p d\mathbf{x} \int \frac{\partial p}{\partial \lambda_j} d\mathbf{x} \big] + \dot{\bar{x}}_j \big[ \int_- \sigma \frac{\partial p}{\partial \bar{x}_j} d\mathbf{x} - \int_- \sigma p d\mathbf{x} \int \frac{\partial p}{\partial \bar{x}_j} d\mathbf{x} \big] \Big\}$$

$$- \dot{\lambda}_2^+ \int_- \sigma p d\mathbf{x} \int_+ \frac{\partial p}{\partial \lambda_2^+} d\mathbf{x} + \dot{\lambda}_2^- \big[ \int_- \sigma \frac{\partial p}{\partial \lambda_2^-} d\mathbf{x} - \int_- \sigma p d\mathbf{x} \int_- \frac{\partial p}{\partial \lambda_2^-} d\mathbf{x} \big] + \int_- \sigma \frac{\partial}{\partial \mathbf{x}} (p\mathbf{S}) d\mathbf{x} = 0. \tag{5.2.7b}$$

Also recall that:

$$\frac{\partial p}{\partial \lambda_j} = -(x_j - \bar{x}_j)^2 p,$$

$$\frac{\partial p}{\partial \bar{x}_j} = 2\lambda_j (x_j - \bar{x}_j) p$$

for $j = 1, 3$, and Eq. (5.2.5a, b). By substituting them into Eq. (5.2.7a, b):

$$\sum_{j=1,3} \Big\{ \dot{\lambda}_j \big[ -\langle \sigma (x_j - \bar{x}_j)^2 \rangle_+ + \langle \sigma \rangle_+ \langle (x_j - \bar{x}_j)^2 \rangle \big] + 2\lambda_j \dot{\bar{x}}_j \big[ \langle \sigma (x_j - \bar{x}_j) \rangle_+ - \langle \sigma \rangle_+ \langle x_j - \bar{x}_j \rangle \big] \Big\}$$

$$+ \dot{\lambda}_2^+ \big[ -\langle \sigma y \rangle_+ + p_+ \langle \sigma \rangle_+ \langle y \rangle_+ \big] - p_- \dot{\lambda}_2^- \langle \sigma \rangle_+ \langle y \rangle_- = \langle \mathbf{S} \cdot \nabla \sigma \rangle_+ + \int \int \sigma p S_y \Big|_{y=0} dx dz / p_+, \tag{5.2.8a}$$

$$\sum_{j=1,3} \Big\{ \dot{\lambda}_j \big[ -\langle \sigma (x_j - \bar{x}_j)^2 \rangle_- + \langle \sigma \rangle_- \langle (x_j - \bar{x}_j)^2 \rangle \big] + 2\lambda_j \dot{\bar{x}}_j \big[ \langle \sigma (x_j - \bar{x}_j) \rangle_- - \langle \sigma \rangle_- \langle x_j - \bar{x}_j \rangle \big] \Big\}$$

$$+ p_+ \dot{\lambda}_2^+ \langle \sigma \rangle_- \langle y \rangle_+ + \dot{\lambda}_2^- \big[ \langle \sigma y \rangle_- - p_- \langle \sigma \rangle_- \langle y \rangle_- \big] = \langle \mathbf{S} \cdot \nabla \sigma \rangle_- - \int \int \sigma p S_y \Big|_{y=0} dx dz / p_-. \tag{5.2.8b}$$

When $\sigma$ depends only on $x$ or $z$, then

$$\langle \sigma y \rangle_{\pm} - p_{\pm} \langle \sigma \rangle_{\pm} \langle y \rangle_{\pm} = 0.$$

(i) $\sigma = x_i - \bar{x}_i$:





Here, we note $\langle\sigma\rangle_+ = \langle\sigma\rangle_- = \langle\sigma\rangle$ as well as

$$\langle x_i - \bar{x}_i\rangle = \langle(x_i - \bar{x}_i)^3\rangle = 0,$$

$$\langle(x_i - \bar{x}_i)^2\rangle - \langle x_i - \bar{x}_i\rangle^2 = \langle(x_i - \bar{x}_i)^2\rangle = 1/2\lambda_i,$$

$$\langle(x_i - \bar{x}_i)^4\rangle = 3/4\lambda_i^2$$

for $i = 1$ and 3. Then, we obtain

$$\dot{\bar{x}}_i = \langle S_i\rangle_+ + \int\int \sigma p S_y\Big|_{y=0}dxdz/p_+, \tag{5.2.9a}$$

$$\dot{\bar{x}}_i = \langle S_i\rangle_- - \int\int \sigma p S_y\Big|_{y=0}dxdz/p_-. \tag{5.2.9b}$$

By taking a weighted sume of them:

$$\dot{\bar{x}}_i = \langle S_i\rangle. \tag{5.2.9c}$$

Eq. (5.2.9c) predicts the evolution of $\bar{x}_i$ with $i = 1$, 3.

(ii) $\sigma = (x_i - \bar{x}_i)^2$: Noting

$$\langle(x_i - \bar{x}_i)^4\rangle - \langle(x_i - \bar{x}_i)^2\rangle^2 = 1/2\lambda_i^2,$$

we obtain

$$\frac{d}{dt}\left(\frac{1}{\lambda_i}\right) - \frac{p_-}{\lambda_i}(\bar{y}_+\dot{\lambda}_2^+ + \bar{y}_-\dot{\lambda}_2^-)$$

$$= 4\langle(x - \bar{x}_i)S_i\rangle_+ + 2\int\int \sigma p S_y\Big|_{y=0}dxdz/p_+, \tag{5.2.10a}$$

$$\frac{d}{dt}\left(\frac{1}{\lambda_i}\right) + \frac{p_+}{\lambda_i}(\bar{y}_+\dot{\lambda}_2^+ + \bar{y}_-\dot{\lambda}_2^-)$$

$$= 4\langle(x - \bar{x}_i)S_i\rangle_- - 2\int\int \sigma p S_y\Big|_{y=0}dxdz/p_- \tag{5.2.10b}$$

for $i = 1$, 3. By taking the weighted sum of the two:

$$\frac{d}{dt}\left(\frac{1}{\lambda_i}\right) = 4\langle(x - \bar{x}_i)S_i\rangle. \tag{5.2.10c}$$

Eq. (5.2.10c) predicts the evolution of $\lambda_i$ with $i = 1$, 3.

Next, note that wehn $\sigma$ depends only on $y$, the sum over $j = 1$, 3 drops out.

(iii) $\sigma = 1$: Both Eqs. (5.2.8a, b) reduce to

$$\dot{\lambda}_2^+/\lambda_2^+ - \dot{\lambda}_2^-/\lambda_2^- = -(\lambda_2^+ + \lambda_2^-)\langle S_y'\rangle, \tag{5.2.11}$$

where

$\langle S_y'\rangle = (r - \bar{z})\bar{x}.$



Note that the system is over–constrained, when the condition with $\sigma = 1$ (*i.e.*, Eq. 3.13) is used along with the results with $\sigma = y$ (*i.e.*, Eq. 3.14c, d). Thus, probably Eq. (5.2.11) is better not counted.

(iv) $\sigma = y$:

$$(2 - p_+)\bar{y}_+^2 \dot{\lambda}_2^+ + p_- \bar{y}_- \bar{y}_+ \dot{\lambda}_2^- = -\langle S_y \rangle' + \bar{y}_+, \tag{5.2.12a}$$

$$p_+ \bar{y}_- \bar{y}_+ \dot{\lambda}_2^+ + (2 - p_-)\bar{y}_-^2 \dot{\lambda}_2^- = \langle S_y \rangle' - \bar{y}_-. \tag{5.2.12b}$$

From Eqs. (3.14a, b), we further obtain:

$$\frac{d}{dt}\left(\frac{1}{\lambda_2^+}\right) = -\frac{1}{\lambda_2^+} + \frac{\langle S_y \rangle'}{2}, \tag{5.2.12c}$$

$$\frac{d}{dt}\left(\frac{1}{\lambda_2^-}\right) = -\frac{1}{\lambda_2^-} - \frac{\langle S_y \rangle'}{2}. \tag{5.2.12d}$$

For a better numerical stability, the time integration is performed in terms of $1/\lambda_2^\pm$ rather than $\lambda_2^\pm$. Note that both $1/\lambda_2^\pm$ present a damping tendency due to the first terms in the right–hand side. Note moreover that asymmetry arising from the second terms, thus when $1/\lambda_2^+$ tends to grow, $1/\lambda_2^-$ tends to decay, and *vice versa*: the general tendency with this pair is qualitatively consistent with a case that the probability to be in positive and negative sides of $y$, $p_\pm$, follow the evolution tendency of the individual phase–space particle. The governing equations for the parameters for the distribution in $x$ and $z$ directions remain unchanged, being constrained by the conditions (i) and (ii) above.

The model II improves from the model I in predicting the $y$–variance (long dash) reasonably (Fig. 9(b)): it grows but with a delay of about 0.5 compared to the actual evolution directly predicted from the Liouville equation. However, as expected, behavior of the variances of the $x$– and $z$–components are as unsuccessful as the case of the model I. The performances of $x$ (solid) and $y$ (long dash) for the mean (Fig. 9(a)) remain the same as for the model I. The failure to predict $\bar{y}$ correctly in spite of a modification of the distribution in $y$ direction is attributed to the fact that the prediction of $\bar{y}_+$ (solid), which decays to zero fairly rapidly after a relatively successful initial prediction until $t \simeq 1$ (Fig. 9 (c)). On the other hand, the prediction of $\bar{y}_-$ (long dash) is reasonable. Here, recall the asymmetry in the initial condition.

When the run is initiated from the origin, the model behavior is even worse: both $\bar{y}_+$ and $\bar{y}_-$ monotoneously decay towards zero with a time scale of $t \simeq 2$ (not shown). Thus, the initial condition symmetric to $y = 0$ somehow worsens the model behavior than improves it, presumaly because the first term in the right–hand sides of Eq. (5.2.12a, b) dominant througout the experiment.

## 5.3 Lorenz (1963) System (Model III)

As a further extension of the model II in the last subsection, we now also constrain the system by $\bar{x}_{pm}$. In this case, $p_1$ furthermore takes the form:

$$p_1 = \begin{cases} p_{10} \exp(-\lambda_1^+ x), & x > 0 \\ p_{10} \exp(\lambda_1^- x), & x < 0 \end{cases} \tag{5.3.1}$$



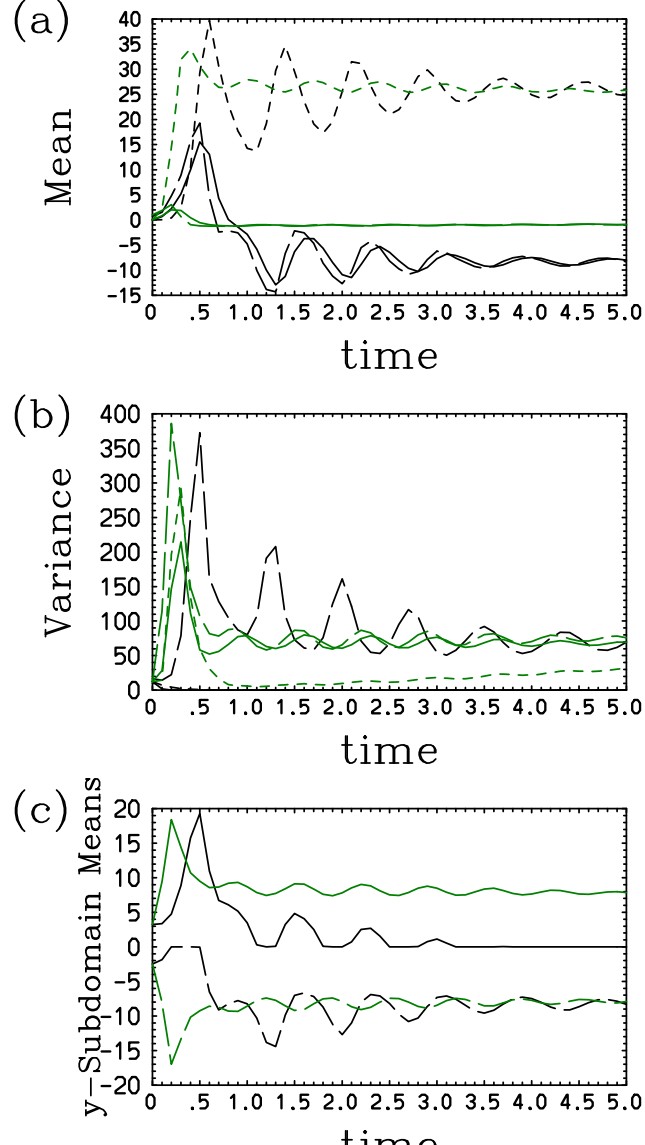

**Figure 9.** Statistics of the Lorenz system with the model II: model results (black), and those by direct prediction with the Liouville equation (green): (a) the means, (b) the variances, (c) $\bar{y}_+$ (solid), and $\bar{y}_-$ (long dash). In the frames (a) and (b), the solid, long–dashed, and short–dashed curves are for $x$–, $y$–, and $z$–components.

Derivation of the equations for the assumed–PDF coefficients also proceeds in a similar manner as in the last subsection, and an only change is that now $\lambda_1^{\pm}$ are predicated by

$$\frac{d}{dt}\left(\frac{1}{\lambda_1^+}\right) = -\frac{P}{\lambda_1^+} + \frac{P}{2}\bar{y},$$  (5.3.2a)

$$\frac{d}{dt}\left(\frac{1}{\lambda_1^-}\right) = -\frac{P}{\lambda_1^-} - \frac{P}{2}\bar{y}.$$  (5.3.2b)





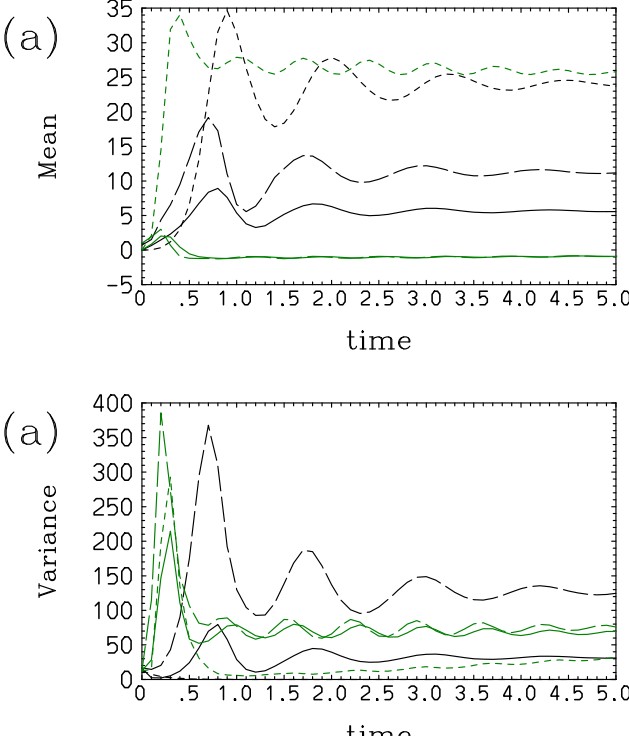

**Figure 10.** Statistics of the Lorenz system with the model III: model results (black), and those by direct prediction with the Liouville equation (green): mean and variance

The major improvement by this modification is a better performance with the $x$–variance (solid: Fig. 10 (b)): it no longer decays out rapidly, but remains about a half of the actual value. On the other hand, the performance of the $z$–variance (short dash: Fig. 10 (b)) does not change. As a rather intriguing modification, both $\bar{x}$ (solid) and $\bar{y}$ (long dash) somehow decided to settle to a positive unstable fixed point this case (Fig. 10(a)).

Even after the attempts with these three assumed PDFs, a key remaining challenge is still to successfully represent the statistics associated with a split of distribution into two peaks. However, as it stands for now, it is not clear what parameter should be added to satisfy this challenge.

## 6   Summary and Discussions

A general methodology for solving the assumed–PDF parameters based on the Liouville equation has been proposed by Yano *et al.* (2024, YLP). The present paper extend this study in several aspects: first, it has been generalized (Sec. 2.2) for the cases that the constraints are defined by limited integral ranges. As a result, the assumed–PDF forms also take different forms over different subdomains, and henceforth, the formulation for the prognostic equations for the PDF parameters have also



been generalized accordingly (Sec. 2.3). Finally, the formulation has been explicitly generalized into multidimensional cases (Secs. 2.4 and 2.5). These further generalized formulations have applied to two simple dynamical system (Secs. 4 and 5, respectively): a convective–energy cycle system proposed by Yano and Plant (2012) as well as Lorenz's (1963) strange–attractor system.

The performances of the assumed–PDF formulations have directly compared with the results from the direct time–integration of the Liouville equation for both systems. It is considered another main originality of the present work: to compare the performance of the assumed–PDF approaches with the results from direct time–integrations of the Liouville equation, and to quantify the resulting errors. Such a rigorous error analysis has not been performed with other PDF formulations found in the literature in my knowledge.

By adopting simple dynamics systems with up to three dependent variables, such a direct time integration becomes feasible. Unfortunately, both testing cases tend to suffer with the same tendencies: regardless of a specific choice, statistics (means and variances) predicted by an assumed–PDF approach gradually deviate from the exact results predicted by the Liouville equation noticeably. Furthermore, some statistics, for example, the sign–dependent conditional averages, $\bar{x}_\pm$ and $\bar{y}_\pm$, in the Lorenz system turn out to be rather difficult to predict properly: in general, only one of the sign–dependent mean pair, $(\bar{x}_\pm, \bar{y}_\pm)$, is

predicted properly, and the other of the pair simply settles to a vanishing value. In this respect, the assumed–PDF approach hardly provide a magic receipt.

Yet, it may also be emphasized that rather difficult cases are taken as test cases: with both systems, the initial Gaussian distribution rapidly evolves into a qualitatively totally different form. Arguably, performances are rather impressive considering the fact that the PDF forms assumed are also qualitatively very different from the actual distributions predicted by the Liouville

equation. The real question still to be to answered is: how well the more standard assumed–PDF approaches perform with the same systems. In that manner, advantage of the present assumed–PDF approach may be better established.

The study has also suggested that the output–constrained distribution principle, proposed in YLP, may not be enough to decide an assumed–PDF form completely. To ensure a good performance of a prediction of the distribution statistics, constraining assumed–PDF forms solely required as outputs for host models, as dictated by this principle, may not be enough. Here, we

have assumed these are to be averages and variances. The present exploratory study has suggested that it is crucial to evaluate the mean evolution of actual forcing terms of a system also accurately, thus they must also be added as a constraint. In both models considered herein, the term $xy$ is crucial in prediction, thus this correlation term must also be properly predicted. It has been shown that adding a constraint of $\langle xy \rangle$ can substantially improve the predictions by assumed–PDF forms, but not necessarily in a satisfactory manner.

Prediction of the PDF of the Lorenz system is inherently difficult due the fact that the solution tends to be clustered around the two unstable fixed points. The assumed–PDF, with all the cases considered so far, have always failed to predict one of those two tendencies under conditional averages, $\bar{x}_\pm$ and $\bar{y}_\pm$. A further possibility to be pursued is to re–initialize a prediction at a middle point, under a spirit of data assimilation, and to examine whether this difficulty is overcome by this procedure.

Some numerical issues have also been revealed by the present study. In some cases, exponential parameters in distribution

can vary to extremes that cause overflow and underflow problems in computations. To avoid this issue, some exponential



parameters have been predicted in their logarithmic form to ensure better numerical stabilities. These parameters must also be bounded both from above and below to avoid overflow and underflow situations, respectively. However, those basic procedures have turned out not to always resolve the stability issues: in the case with the model III for the Lorenz system (Sec. 5.3), a rather small nondimensional time step of $10^{-4}$ has been necessary to run a prediction long enough, but it still ultimately crushes. The numerical stability of this case must more closely be investigated in its own right.

As a whole, the present study reveals inherent difficulties of predicting a distribution accurately only with a limited number of distribution parameters. Especially, we have faced a universal tendency of the variance of the distribution decays with time, when it must increase, regardless of the choice of an assumed distribution form.

This behavior is reminiscence of the variance collapse identified as a major problem in data assimilation when it is performed with an ensemble. In the latter case, it is often found that the probability weight assigned to each ensemble member collapses close to zero saved for a single member close to the average (*i.e.*, weight collapse: van Leeuwen 2003, Poterjoy 2016), and henceforth, the prior estimate of the variance also collapses as a consequence (*cf.*, Snyder *et al.* 2008): see van Leeuwen (2009) for more backgrounds. Though the cause of this problem is usually attributed to a relatively small ensemble size and their tendency to collapse into a "stable" state, the present study suggests that this tendency is more universal, and it can happen whenever a highly truncated representation of a distribution is adopted, even with an approach of considering a full dimension of the phase space as in the present case.

Those ensemble–based assimilation studies, in turn, attempt various remedies to alleviate this tendency: the simplest is to inflate the variance by a certain factor with time to prevent,t its collapse (Anderson and Anderson 1999, Anderson 2001). More generally, resampling approaches can, at least, partially delay the collapse of the weights (Snyder *et al.* 2008, Anderson 2001). However, it appears that neither of the approaches is directly applicable to the assumed–PDF formulation in the present study. First, the inflation is nothing other than adding an extra adjustable parameter, which can be chosen only when an exact PDF evolution is known *a priori*. Resampling approaches do not work simply by not adopting an ensemble formulation.

The most feasible solution to solve the variance collapse under the assume–PDF approaches would be to include a feedback of the truncated distribution parameters in a form of a parameterization, in a similar manner as effects of higher moments are represented by certain hypothesis in the turbulence–closure models (Mellor 1973, Mellor and Yamada 1974). However, parameterizations of the higher–order assumed PDF parameters are completely new frontiers, to which much investments are required before we can propose any specific solution.

*Code availability.* The fotran codes used in the present study are available by request to the author.





## Appendix: Integrals of Two–Dimensional Gaussian Distribution

### A.1 Normalization

The normalization condition for a distribution with two variables, $(x, y)$, is given by

$$\int_{-\infty}^{+\infty} \int_{-\infty}^{+\infty} p \, dx \, dy = 1.$$

Recall a Gaussian integral formula:

$$\int_{-\infty}^{+\infty} e^{-\lambda \xi^2} d\xi = \left(\frac{\pi}{\lambda}\right)^{1/2}.$$

For casting the integral in $x$ in this form, we re-arrange a part of the exponent as follows:

$$\lambda_1(x - \bar{x})^2 + \lambda_3(x - \bar{x})(y - \bar{y})$$
$$= \lambda_1[x - \{\bar{x} - \frac{\lambda_3}{2\lambda_1}(y - \bar{y})\}]^2 - \lambda_1[\bar{x} - \frac{\lambda_3}{2\lambda_1}(y - \bar{y})]^2 + \lambda_1\bar{x}^2 - \lambda_3(y - \bar{y})\bar{x}$$

Thus,

$$\int_{-\infty}^{+\infty} \frac{p}{p_0} dx = \left(\frac{\pi}{\lambda_1}\right)^{1/2} \exp[-\lambda_2(y - \bar{y})^2 + \lambda_1\{\bar{x} - \frac{\lambda_3}{2\lambda_1}(y - \bar{y})\}^2 - \lambda_1\bar{x}^2 + \lambda_3(y - \bar{y})\bar{x}] \tag{A.1}$$

The remaining exponent can be re-arranged as

$$\lambda_2(y - \bar{y})^2 - \lambda_1[\bar{x} - \frac{\lambda_3}{2\lambda_1}(y - \bar{y})]^2 + \lambda_1\bar{x}^2 - \lambda_3(y - \bar{y})\bar{x} = (\lambda_2 + \frac{\lambda_3^2}{4\lambda_1})(y - \bar{y})^2$$

Thus a further integral of (A.1) in $y$ leads to

$$\int_{-\infty}^{+\infty} \int_{-\infty}^{+\infty} \frac{p}{p_0} dx \, dy = \left(\frac{\pi}{\lambda_1}\right)^{1/2} \left(\frac{\pi}{\lambda_2 + \lambda_3^2/4\lambda_1}\right)^{1/2} = \frac{\pi}{(\lambda_1\lambda_2 + \lambda_3^2/4)^{1/2}}$$

The final expression proves Eq. (4.1.2a, b, c).

### A.2 Moments

The moments given by Eq. (3.5a–j) are derived by using relationships obtained by taking the differentiation of the Gaussian distribution in recursive manner:

$$\frac{\partial}{\partial x}[e^{-\{\lambda_1(x - \bar{x})^2 + \lambda_3(x - \bar{x})(y - \bar{y})\}}] = [2\lambda_1(x - \bar{x}) + \lambda_3(y - \bar{y})]e^{-[\lambda_1(x - \bar{x})^2 + \lambda_3(x - \bar{x})(y - \bar{y})]}.$$



The integral of the right hand side leads to

$$\int\limits_{-\infty}^{+\infty} [2\lambda_1(x-\bar{x}) + \lambda_3(y-\bar{y})]e^{-[\lambda_1(x-\bar{x})^2 + \lambda_3(x-\bar{x})(y-\bar{y})]}dx$$

$$= -e^{-[\lambda_1(x-\bar{x})^2 + \lambda_3(x-\bar{x})(y-\bar{y})]}|_{-\infty}^{+\infty} = 0. \tag{A.2}$$

This relation immediately finds:

$$2\lambda_1\langle x-\bar{x}\rangle + \lambda_3\langle y-\bar{y}\rangle = 0. \tag{A.3a}$$

Same wise, by symmetry:

$$2\lambda_2\langle y-\bar{y}\rangle + \lambda_3\langle x-\bar{x}\rangle = 0. \tag{A.3b}$$

Solving them together leads to Eq. (3.5a) provided $4\lambda_1\lambda_2 - \lambda_3^2 \neq 0$.

To generalize a type of relations like (A.3a, b), we note that Eq. (A.2) is still valid by multiplying a weight, $\sigma$, with any function of $y$. Letting $\sigma = y - \bar{y}$:

$$2\lambda_1\langle (x-\bar{x})(y-\bar{y})\rangle + \lambda_3\langle (y-\bar{y})^2\rangle = 0, \tag{A.4a}$$

and by symmetry,

$$2\lambda_2\langle (x-\bar{x})(y-\bar{y})\rangle + \lambda_3\langle (x-\bar{x})^2\rangle = 0. \tag{A.4b}$$

By combining them together, we obtain the relations:

$$\langle (x-\bar{x})(y-\bar{y})\rangle = -\frac{\lambda_3}{2\lambda_2}\langle (x-\bar{x})^2\rangle = -\frac{\lambda_3}{2\lambda_1}\langle (y-\bar{y})^2\rangle, \tag{A.5a}$$

$$\langle (x-\bar{x})^2\rangle = -\frac{2\lambda_2}{\lambda_3}\langle (x-\bar{x})(y-\bar{y})\rangle, \tag{A.5b}$$

$$\langle (y-\bar{y})^2\rangle = -\frac{2\lambda_1}{\lambda_3}\langle (x-\bar{x})(y-\bar{y})\rangle. \tag{A.5c}$$

Eq. (A.2) can further be used for deriving similar expressions for higher-moment integrals

$$\int\limits_{-\infty}^{+\infty} [2\lambda_1(x-\bar{x}) + \lambda_3(y-\bar{y})]^n e^{-[\lambda_1(x-\bar{x})^2 + \lambda_3(x-\bar{x})(y-\bar{y})]}dx \tag{A.6}$$

with $n$ an arbitrary integral. We first set $n = 2$, and obtain by a partial integral:

$$\int\limits_{-\infty}^{+\infty} [2\lambda_1(x-\bar{x}) + \lambda_3(y-\bar{y})]^2 e^{-[\lambda_1(x-\bar{x})^2 + \lambda_3(x-\bar{x})(y-\bar{y})]}dx$$

$$= -[2\lambda_1(x-\bar{x}) + \lambda_3(y-\bar{y})]e^{-[\lambda_1(x-\bar{x})^2 + \lambda_3(x-\bar{x})(y-\bar{y})]}|_{-\infty}^{+\infty}$$

$$+ 2\lambda_1 \int\limits_{-\infty}^{+\infty} e^{-[\lambda_1(x-\bar{x})^2 + \lambda_3(x-\bar{x})(y-\bar{y})]}dx,$$





thus

$$\langle [2\lambda_1(x-\bar{x})+\lambda_3(y-\bar{y})]^2 \rangle = -\int_{-\infty}^{+\infty}[2\lambda_1(x-\bar{x})+\lambda_3(y-\bar{y})]p|_{-\infty}^{+\infty}dy + 2\lambda_1.$$

Since the partial integral vanishes, by expanding the left-hand side, we obtain

$$4\lambda_1^2\langle(x-\bar{x})^2\rangle + \lambda_3^2\langle(y-\bar{y})^2\rangle + 4\lambda_1\lambda_3\langle(x-\bar{x})(y-\bar{y})\rangle = 2\lambda_1. \tag{A.7a}$$

By symmetry, we also obtain

$$4\lambda_2^2\langle(y-\bar{y})^2\rangle + \lambda_3^2\langle(x-\bar{x})^2\rangle + 4\lambda_2\lambda_3\langle(x-\bar{x})(y-\bar{y})\rangle = 2\lambda_2. \tag{A.7b}$$

By substituting Eqs. (A.5b, c) into the above, we obtain Eq. (3.5d). Its substitution back to Eqs. (A.5b, c), respectively, lead to Eqs. (3.5b, c).

To obtain the expressions for the third moments, we now set $n=3$ in Eq. (A.6). Since this whole integral vanishes, by expanding this integral, we obtain:

$$8\lambda_1^3\langle(x-\bar{x})^3\rangle + \lambda_3^3\langle(y-\bar{y})^3\rangle + 12\lambda_1^2\lambda_3\langle(x-\bar{x})^2(y-\bar{y})\rangle + 6\lambda_1\lambda_3^2\langle(x-\bar{x})(y-\bar{y})^2\rangle = 0, \tag{A.8a}$$

and also by symmetry,

$$8\lambda_2^3\langle(y-\bar{y})^3\rangle + \lambda_3^3\langle(x-\bar{x})^3\rangle + 12\lambda_2^2\lambda_3\langle(x-\bar{x})(y-\bar{y})^2\rangle + 6\lambda_2\lambda_3^2\langle(x-\bar{x})^2(y-\bar{y})\rangle = 0. \tag{A.8b}$$

As an extension of (A.3a, b), we obtain:

$$2\lambda_1\langle(x-\bar{x})(y-\bar{y})^2\rangle + \lambda_3\langle(y-\bar{y})^3\rangle = 0,$$
$$2\lambda_2\langle(x-\bar{x})^2(y-\bar{y})\rangle + \lambda_3\langle(x-\bar{x})^3\rangle = 0.$$

They lead to:

$$\langle(x-\bar{x})^3\rangle = -\frac{2\lambda_2}{\lambda_3}\langle(x-\bar{x})^2(y-\bar{y})\rangle, \tag{A.9a}$$

$$\langle(y-\bar{y})^3\rangle = -\frac{2\lambda_1}{\lambda_3}\langle(x-\bar{x})(y-\bar{y})^2\rangle. \tag{A.9b}$$

By substituting Eqs. (A.9a, b) into Eqs. (A.8a, b), we obtain

$$\lambda_1(4\lambda_1\lambda_2-3\lambda_3^2)\langle(x-\bar{x})^2(y-\bar{y})\rangle - \lambda_3^3\langle(x-\bar{x})(y-\bar{y})^2\rangle = 0$$
$$\lambda_2(4\lambda_1\lambda_2-3\lambda_3)\langle(x-\bar{x})(y-\bar{y})^2\rangle - \lambda_3^3\langle(x-\bar{x})^2(y-\bar{y})\rangle = 0$$

By solving them for $\langle(x-\bar{x})^2(y-\bar{y})\rangle$ and $\langle(x-\bar{x})(y-\bar{y})^2\rangle$, and also substituting this result into (A.9a, b), we obtain Eq. (3.5e).





Finally, for obtaining the results for the fourth moments, we set $n = 4$ in Eq. (A.6), which leads to:

$$16\lambda_1^4\langle(x-\bar{x})^4\rangle + \lambda_3^4\langle(y-\bar{y})^4\rangle + 32\lambda_1^3\lambda_3\langle(x-\bar{x})^3(y-\bar{y})\rangle$$
$$+8\lambda_1\lambda_3^3\langle(x-\bar{x})(y-\bar{y})^3\rangle + 24\lambda_1^2\lambda_3^2\langle(x-\bar{x})^2(y-\bar{y})^2\rangle = 12\lambda_1^2,$$

(A.10a)

and also by symmetry,

$$16\lambda_2^4\langle(y-\bar{y})^4\rangle + \lambda_3^4\langle(x-\bar{x})^4\rangle + 32\lambda_2^3\lambda_3\langle(x-\bar{x})(y-\bar{y})^3\rangle$$
$$+8\lambda_2\lambda_3^3\langle(x-\bar{x})^3(y-\bar{y})\rangle + 24\lambda_2^2\lambda_3^2\langle(x-\bar{x})^2(y-\bar{y})^2\rangle = 12\lambda_2^2.$$

(A.10b)

As an extension of Eqs. (A.3a, b), we obtain

$$2\lambda_1\langle(x-\bar{x})(y-\bar{y})^3\rangle + \lambda_3\langle(y-\bar{y})^4\rangle = 0,$$
$$2\lambda_2\langle(x-\bar{x})^3(y-\bar{y})\rangle + \lambda_3\langle(x-\bar{x})^4\rangle = 0,$$

which lead to

$$\langle(x-\bar{x})(y-\bar{y})^3\rangle = -\frac{\lambda_3}{2\lambda_1}\langle(y-\bar{y})^4\rangle,$$ (A.11a)

$$\langle(x-\bar{x})^3(y-\bar{y})\rangle = -\frac{\lambda_3}{2\lambda_2}\langle(x-\bar{x})^4\rangle.$$ (A.11b)

The results of Eqs. (A.7a, b) are extended by applying weights of $(y-\bar{y})^2$ and $(x-\bar{x})^2$, respectively, and we obtain:

$$4\lambda_1^2\langle(x-\bar{x})^2(y-\bar{y})^2\rangle + \lambda_3^2\langle(y-\bar{y})^4\rangle + 4\lambda_1\lambda_3\langle(x-\bar{x})(y-\bar{y})^3\rangle = \frac{\lambda_1}{\lambda_2}(1-\frac{\kappa}{4})^{-1}$$

$$4\lambda_2^2\langle(x-\bar{x})^2(y-\bar{y})^2\rangle + \lambda_3^2\langle(x-\bar{x})^4\rangle + 4\lambda_2\lambda_3\langle(x-\bar{x})^3(y-\bar{y})\rangle = \frac{\lambda_2}{\lambda_1}(1-\frac{\kappa}{4})^{-1}$$

By substituting Eqs. (A.11a, b) into the above, we obtain

$$\langle(x-\bar{x})^4\rangle = \frac{4\lambda_2^2}{\lambda_3^2}\langle(x-\bar{x})^2(y-\bar{y})^2\rangle - \frac{\lambda_2}{\lambda_1\lambda_3^2}(1-\frac{\kappa}{4})^{-1},$$ (A.12a)

$$\langle(y-\bar{y})^4\rangle = \frac{4\lambda_1^2}{\lambda_3^2}\langle(x-\bar{x})^2(y-\bar{y})^2\rangle - \frac{\lambda_1}{\lambda_2\lambda_3^2}(1-\frac{\kappa}{4})^{-1},$$ (A.12b)

We first substitute Eqs. (A.11a, b) into Eqs. (A.10a, b), then substitute Eqs. (A.12a, b) into the latter. This gives Eq. (3.5h). Substituting back Eq. (3.5h) into (A.12a, b) gives Eqs. (3.5f, g), and further substitutions of them into Eqs. (A.11a, b) lead to Eqs. (3.5i, j).

*Author contributions.*   The present work is entirely due to the present author.



*Competing interests.* I declare no competing interests.

*Acknowledgements.* The present work has evolved over years under discussions with various people, but extensive discussions with Vince Larson and Vaughan Phillips are especially acknowledged.



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
