# Peer review of "Prognostic Assumed-PDF (DDF) Approach: Further Generalization and Demonstrations"

_EGUsphere, 2024_

## Author Comment (AC1)

**Response to the Reviewer RC1**

Please note that in the following response, the Review texts are quoted by »...«.

I much appreciate the critical, constructive comments by the present Reviewer. Yet, first, I have a major request for the clarification:

**The Major Request to the Present Reviewer:**

I read the following paragraph to be the main summary of the Review:

»the structure of the paper could be modified to improve the clarity of the discussion. In particular, some choices are not adequately explained. Adding some references between sections would be helpful to improve the coherence of the paper.«

Unfortunately, I cannot understand how and in what manner the "structure of the paper" must be modified. The Reviewer indeed tries to paraphrase the matter by the two sentences that follow. However, it is not clear how these issues to be reflected upon the "modification of the structure" of the paper. As far as I can tell, all these issues can be solved by elaborating texts at given places without modifying the structure of the paper.

Fortunately, this article is in interactive discussion phase. Thus, I would much appreciate it, if the Reviewer could elaborate on this comment online. This elaboration would be crucial for me to properly revise the manuscript by following the Reviewer's recommendation of the »major revision«, because this very issue is clearly the "major issue" to be addressed.

**Summary of the Response:**

I emphasize that the present Reviewer does not point to any defect in the methodology adopted in this study. Considering the originality of the methodology, thus, the present manuscript should be accepted for the publication. I frankly admit that the outcome from this methodology (so far) is not a total success. However, the results should be made public for the further common investigations for the progress, rather than bringing them back to my personal drawer.

**Lead Paragraph:**

I am glad to read that the present Reviewer has followed the »main message« of the present work very well, as seen in the summary given in the beginning. To quote in full:

»This paper investigates a method to predict the evolution of the parameters of assumed probability density functions (PDF), which is applied to different dynamical systems for which an exact solution of the Liouville equation is available. In this paper the method is extended to cases in which constraints are defined over subdomains, the distribution takes different forms in different subdomains and to multidimensional cases.«

Thus, it is rather difficult for me to understand what the Reviewer asks about the »main message« of the present study.

Furthermore, I find it rather unfortunate that the present Reviewer bluntly conclude that »the method fails«: I am more than happy to admit the working of this method is far from perfect. However, as emphasized in the concluding section (L547–550), the present study is attempting a difficult task, most likely next to impossible, of predicting evolution of a distribution only with a limited number of parameters, but in a consistent manner.

Here, the so–called assumed PDF approach already exists for a long time. Yet, this work is original in attempting to predict the evolution of a distribution in a self–consistent manner, and verify the performance taking simply dynamical systems. Such an effort does not exist in the literature, because the existing assumed–PDF schemes are developed case by case with *ad hoc* closures, without a generality. Thus, it is simply not possible to perform the verifications of those schemes by taking simple dynamical systems.

**General comments:**

In responding to the General Comments, I would first emphasize that the present work is a sequel to the first paper (YLP) under review to ACP, which is also available online. Thus, I've been assuming that any readers, including the Reviewers, would read this ACP paper first before reading this manuscript. For this reason, the introduction only presents the main issues in a very succinct manner, leaving the full discussions to the ACP paper. Especially extensive references are already found in the latter paper. This very simply point will be explicitly emphasized in the final manuscript.

It also follows that the »motivation« of the present paper must also be obvious: only the simplest application is presented in the first paper in ACP. Thus, it must naturally be tested more extensively.

In my own opinion, the abstract and the introduction are already presented in clear and succinct manners: the main result is an inherent difficulty of properly predicting the variances by using only a limited number of PDF parameters, as clearly stated in the abstract.

Yet, I also note that the present version is still too terse to attain even a minimum self–contained reading. Thus, in revision, some further elaborations will be attempted, some of them also remarked in the following.

The specific choices for the weights, $\sigma_l$, as well as the assumed PDF form follow the output–controlled distribution principle proposed in YLP. This point as well as the basic idea of this principle will be better emphasized in the final manuscript. See the response to L195–196 below for more.

Finally, the Reviewer asks me for »adding some references between sections«. Yet, to perform this modification, I would need to know what the Reviewer means by that more precisely.

**Specific comments:**

L4-6: The following sentence will be added to the abstract (if the word number limit permits) in revision for clarify the context better: "The general formulation

developed here is applicable to a wide range of the problems, including the frequency distributions of subgrid–scale variables, hydrometeor size distributions, as well as to probability distributions characterizing data uncertainties."

L8: "the common cause" will be replaced by "a common cause ... due to low dimensionality" in revision to be grammatically correct, and also for making it clear that an exact cause is not known, but also with an addition of the phrase suggesting the required key condition.

L17: The reference to Yano *et al.* (2018, BAMS) will be added here to allude to the weather forecasting, as suggested by the Reviewer, although this reference is already found in YLP.

L50: Meaning of $\sigma_l$: Its basic »meaning« must be clear from the phrase that introduces $\sigma_l$: "weighting it by $\sigma_l$". Thus, it is a weight. Eq. (2.1.7) and the following discussions should further clarify the »meaning« of $\sigma_l$. Please note that a more careful derivation and the discussions are found in Sec. 5.1 of YLP, as will be explicitly remarked in revision.

Here, please understand that the presentation here is extremely terse, because all these details are more carefully discussed already in YLP: the readers should refer to it to understand those details. I believe that this point is already implied in the manuscript (*e.g.*, L23, L35), but it will be made more explicit in revision both in the introduction as well as in the beginning of Sec. 2.

L104: A brief description of the output–controlled distribution principle proposed in YLP: the essence of this principle is already introduced in L55–56: "YLP suggest to choose those constraints to be the outputs that are required in a host model." In revision, this sentence will be immediately followed by "and call it the output–controlled distribution principle."

L109: As Eq. (2.2.3) demonstrates, when different constraints are introduced in two subdomains, different forms of distributions are predicted from the maximum entropy principle. Here, the two subdomains are separated by the sign of $x$ (or $\phi$), thus the two different distribution forms must be assumed over those two subdomains, as shown in the definition that immediately follows L109. These points will be made more explicit in revision. Here, I also realize that the original presentation was slightly out of order: in revision, the paragraph here will be re–structured to a better order.

Similar remarks will also be added in the beginning of Sec. 2.5 to suggest in what case we further need to introduced different distribution forms also depending on the sign of $y$. Again, note a slight disorder in presentation here, which will be amended in revision.

Section 3: Visualization of the distribution: here, I assume that the Reviewer is asking for a visualization of the distribution defined by Eq. (3.1). However, this is simply a two–dimensional Gaussian distribution, whose form must be widely known. In fact, the use of the multi–dimensional Gaussian distribution is fairly common in the assumed–PDF literature (*e.g.*, Golaz *et al.* 2002, Larson and Golaz 2005). For this reason, instead of visualizations, those references will be simply added in revision.

L195-196: Choice of $\sigma$: the weights, $\sigma_l$, are chosen as means and variances throughout the paper, because these are the simplest quantities required as outputs. This basic point will be remarked in an earlier part of the manuscript in revision in association with Eq. (2.1.7).

L255: Choice of $\sigma$: please refer to my response to L195–196 just above.

L293-294: A possible solution to improve the solution with an assumed PDF: the reason for the failure to capture the basic evolution of the distribution is discussed in the paragraph (L295–300) that immediately follows. Alternative distribution forms are considered in Secs. 4.2 and 4.3 in order to overcome this defect, as clearly stated in the beginning of each section (L305, L343).

L308: An integral over an infinite domain at each time step, as required, is substantially more expensive numerically than just predicting a few PDF parameters. Obviously a sum of far more than few points is required to obtain an integral with an acceptable accuracy. This computation cost, to be performed at every time step, is much more substantial compared to the cost for simply integrating only few parameters in time, as the present method is designed to do. This elaboration will be added in revision.

L311: "Minor disadvantage": based on the way that the Reviewer is asking the question, it is clear that the Reviewer understands a "disadvantage" that "a distribution can spread to $x < 0$": »the solution covers unphysical values«. Yet, this is only a "minor disadvantage" for the reason explained by the sentence that immediately follows (L312–313).

L348: Purpose of section 4.3: the purpose of Sec. 4.3 is to seek "an alternative possibility" *to Sec. 4.2*. In revision, this additional phrase will be added for a further clarity. This method was actually formulated explicitly, but not presented in detail, because as remarked, no substantial difference from the case with Sec. 4.2 was found. I believe it worthwhile to remark on this unsuccessful alternative attempt, even just briefly, because it is a very natural choice to try.

L540: »This study applies the assumed-PDF approach to dynamical systems for which it is possible to compute the solution of the Liouville equation, but the method generally fails to reproduce the exact solution. What would be the appropriate procedure for cases where the exact solution is not available?«: I do not know the answer to this question. However, it seems to me that it is more constructive to seek better assumed–PDF forms that works better for simple dynamical systems. Then, a similar approach can also be applied to more complex systems, where the direct verifications by the Liouville equation is not feasible.

L566-567: The sentence redundant?: The sentence of concern will be removed in revision.

**Technical corrections:**

L70: The superscript $+$ in $\lambda_l$ in the second integral: thank you for pointing out a typo here. The superscript $+$ here will be corrected to $-$ in revision.

Additional technical corrections suggested by the present Reviewer for L134, Caption in Figs. 3 and 4, L481, L527, L530, L532, L535, and L583 will be

performed in revision.

---

## Author Comment (AC2)

**Response to the Reviewer RC2**

I much appreciate the careful balanced thoughts on this manuscript by the present Reviewer.

Here, I emphasize that the present Reviewer does not point to any defect in the methodology adopted in this study. Considering the originality of the methodology, thus, the present manuscript should be accepted for the publication. I frankly admit that the results from this methodology are not a total success. However, the results should be made public for the further common investigations for the progress, rather than bringing them back to my personal drawer.

For this goal, I would also appreciate it, if the Reviewer could elaborate on the matters not clear for me in the following, best, in the interactive mode.

Please note that in the following response, the Review texts are quoted by »...«.

I much appreciate a positive evaluation by the present Reviewer stating that »the manuscript provides a careful and well-done analytical analysis of the Liouville equations and the assumed Pdf approach.« Yet, at the same time, the Reviewer expresses a strong reservation, remarking that it »likely of rather limited practical benefit as also noted by the author.« Unfortunately, I do not understand what the Reviewer means by »limited practical benefit«. Although the Reviewer says that I also note it, the phrase itself is not of mine.

In response, I emphasize the fact that a very solid analysis of the performance is performed in the present study by directly comparing the assumed–PDF results with direct numerical results with the Liouville equation by taking some simple dynamical systems. Providing a general, robust formulation that enables this kind of comparisons is a real originality of the present study: this is not possible with the current existing assumed–PDF schemes, because these are formulated only in a case–by–case manner with only specific applications in mind. Thus the performance of those schemes cannot be tested by simple dynamical systems.

The robustness of the proposed formulation is discussed extensively in YLP (submitted to ACP), which is also currently available online. The goal of the present manuscript is, as clearly acknowledged by the present Reviewer, to test this formulation for more advanced cases.

One may judge that the obtained results are not quite promising, and even a failure. However, one must also count on the basic fact that the assumed PDF attempts something almost impossible: to perform an accurate prediction of a distribution only by using a limited number of parameters. For this reason, one should consider the obtained results are important demonstration of the fundamental difficulties with the assumed–PDF methods in general, not only with a particular approach adopted herein..

All these points will be more extensively discussed in the final manuscript.

-Abstract and Conclusion: Here, the Reviewer objects my interpretation that the tendency for vanishing variance with the present approach is "likely a common cause of collapse in variance found in ensemble-based data assimilation". I do not know in what sense »This is not founded here and partly misleading«:

it is clearly a common feature that we can identify both with the ensemble–based data assimilation and the present study. Here, I am afraid, the Reviewer slightly misunderstands the standard procedures with ensemble–based data assimilation: as the present Reviewer correctly points out, indeed, they typically »introduce spread/variance through perturbations to the observations and the model« (*cf.*, L582–584). However, this procedure is necessary, because otherwise, the variance would collapse (*cf.*, L574–581), as found in the present study. Somehow the present Reviewer misses this latter point. Please refer to L574–584 of the manuscript.

- Analysis: I am glad to know that the present Reviewer finds that »the analysis is accurately done«.

-The present Reviewer clearly acknowledges that the present manuscript is an extension of another manuscript currently available as a »Technical Memorandum« in EGUsphere (submitted to ACP). The Reviewer states explicitly that »I liked the example/solution part«. Nevertheless, also adds »too long without benefit«, however, with no specific reason provided. More specifically, the Reviewer suggest to drop Secs. 5.2 and 5.3: this suggestion is just odd, because these two subsections pursue the alternative possibilities that partially overcome the defects found with the first model considered in Sec. 5.1. Inclusion of these two subsections is crucial for this reason.

**Typos:**

-l12 "as in values themselves": will be modified to "as frequency distributions of variables at a single macroscopic point"

Further typos at l24, l46, and l224, and l287 will also be corrected, as well as the errors in the labeling of Figures 7 and 8.

---

## Author Response (AR1)

**Response to the Reviewers**

**General Remarks**

The following response is essentially identical to what I responded to each Reviewer in the interactive discussion phase, apart from the change of the future tense *e.g.*, "will be modified" to the present complete tense *e.g.*, "have been modified", as well as minor edit. Although I was hoping that the Reviewer may provide me further elaborations on unclear part of the original comments, unfortunately I received none. For this reason, no further action has been made on the comments that I could not follow since the time of the interactive discussion phase.

Please note that in the following response, the Review texts are quoted by »...«.

**Summary**

It is emphasized that neither of the Reviewers points to any defect in the methodology adopted in this study. Considering the originality of the methodology, thus, the present manuscript should be accepted for the publication. I frankly admit that the results from this methodology are not a total success. Yet, I strongly believe that the results should be made public for the further common investigations for the progress, rather than bringing them back to my personal drawer.

It is also emphasized that the general framework of the *assumed PDF* is not my own "invention". Rather this is the most basic framework adopted in many of the distribution studies in the all three disciplines (subgird, microphysics, assimilation), as emphasized in revision (L560–566).

Uniqueness of the present study, as an extension of YLP, is to propose a methodology that can directly predict the PDF parameters with time, rather than diagnosing them from other statistical information (typically from the moments), and compare the accuracy of the method with direct computations by the Liouville equation. If the Reviewers consider the present study to be failure, this conclusion must also be applied to the whole assumed–PDF based studies, because in the current literature, such a direct comparison simply does not exist.

Thus, the present results have a wide range of implications to the current modeling of the "distributions" in general.

**Response to the Reviewer RC1**

Please refer to the General Remarks and the Summary in the beginning of the file before reading to the more specific response to the Reviewer RC1 here.

I much appreciate the critical, constructive comments by the present Reviewer.

**Lead Paragraph:**

I am glad to read that the present Reviewer has followed the »main message« of the present work very well, as seen in the summary given in the beginning.

To quote in full:

»This paper investigates a method to predict the evolution of the parameters of assumed probability density functions (PDF), which is applied to different dynamical systems for which an exact solution of the Liouville equation is available. In this paper the method is extended to cases in which constraints are defined over subdomains, the distribution takes different forms in different subdomains and to multidimensional cases.«

Thus, it is rather difficult for me to understand what the Reviewer asks about the »main message« of the present study.

Furthermore, I find it rather unfortunate that the present Reviewer bluntly conclude that »the method fails«: I am more than happy to admit the working of this method is far from perfect. However, as emphasized in the concluding section (L547–550: L575–583), the present study is attempting a difficult task, most likely next to impossible, of predicting evolution of a distribution only with a limited number of parameters, but in a consistent manner.

Here, the so–called assumed PDF approach already exists for a long time. Yet, this work is original in attempting to predict the evolution of a distribution in a self–consistent manner, and verify the performance by taking simple dynamical systems. Such an effort does not exist in the literature, because the existing assumed–PDF schemes are developed case by case with *ad hoc* closures, without a generality. Thus, it is simply not possible to perform the verifications of those schemes by taking simple dynamical systems (*cf.*, L560–566).

**General comments:**

In responding to the General Comments, I would first emphasize that the present work is a sequel to the first paper (YLP) under review to ACP, which is also available online. Thus, I've been assuming that any readers, including the Reviewers, would read this ACP paper first before reading this manuscript. For this reason, the introduction only presents the main issues in a very succinct manner, leaving the full discussions to the ACP paper. Especially extensive references are already found in the latter paper. These very simple points have been explicitly emphasized in the final manuscript (L35–36, L48).

It also follows that the »motivation« of the present paper must also be obvious: only the simplest application is presented in the first paper in ACP. Thus, it must naturally be tested more extensively.

In my own opinion, the abstract and the introduction are already presented in clear and succinct manners: the main result is an inherent difficulty of properly predicting the variances by using only a limited number of PDF parameters, as clearly stated in the abstract.

Yet, I also note that the original version was still too terse to attain even a minimum self–contained reading. Thus, in revision, some further elaborations have been attempted, some of them also remarked in the following.

Some choices: The specific choices for the weights, $\sigma_l$, as well as the assumed PDF form follow the output–controlled distribution principle proposed in YLP.

This point as well as the basic idea of this principle have been better emphasized in the final manuscript. See the response to L195–196 below for more.

Furthermore, the Reviewer asks me for »adding some references between sections«. Yet, to perform this modification, I would need to know what the Reviewer means by that more precisely.

As I read, the main suggestion of the present Reviewer is to modify »the structure of the paper«. However, unfortunately, I do not read any remarks by the present Reviewer how and in what manner the »structure of the paper« must be modified. The Reviewer indeed tries to paraphrase the matter by the two sentences that follow. However, the given paraphrasing only points to the issues, that can be solved by elaborating texts at given places without modifying the structure of the paper, as already responded above.

Although the present Reviewer recommends the »major revision«, unfortunately, I could not modify »the structure of the paper« in any major manner for the reason just explained.

**Specific comments:**

L4-6: The following sentence has been added to the abstract: "The general formulation developed here is applicable to a wide range of the problems, including the frequency distributions of subgrid–scale variables, hydrometeor size distributions, as well as to probability distributions characterizing data uncertainties." (L6–18)

L8 (L10–11 in revision): "the common cause" has been replaced by "a common cause . . . due to the low dimensionality" in revision to be grammatically correct, and also for making it clear that an exact cause is not yet known, but also with an addition of the phrase suggesting the required key condition.

L17 (L19–20 in revision): The reference to Yano *et al.* (2018, BAMS) has been added here to allude to the weather forecasting, as suggested by the Reviewer, although this reference is already found in YLP.

L50 (L64 in revision): Meaning of $\sigma_l$: Its basic »meaning« must be clear from the phrase that introduces $\sigma_l$: "weighting it by $\sigma_l$". Thus, it is a weight. Eq. (2.1.7) and the following discussions should further clarify the »meaning« of $\sigma_l$. Please note that a more careful derivation and the discussions are found in Sec. 5.1 of YLP, as have been explicitly remarked in revision (L72–73).

Here, please understand that the presentation here is extremely terse, because all these details are more carefully discussed already in YLP: the readers should refer to it to understand those details. I believe that this point is already implied in the original manuscript (*e.g.*, L23, L35), but it has been made more explicit in revision both in the introduction (L35–36) as well as in the beginning of Sec. 2 (L48).

L104 (L126 in revision): A brief description of the output–controlled distribution principle proposed in YLP: the essence of this principle is already introduced in L55–56 (L69–70 in revision): "YLP suggest to choose those constraints to be the outputs that are required in a host model." In revision, this sentence has

been immediately followed by "and call it the output–controlled distribution principle."

L109 (L135 in revision): As Eq. (2.2.3) demonstrates, when different constraints are introduced in two subdomains, different forms of distributions are predicted from the maximum entropy principle. Here, the two subdomains are separated by the sign of $x$ (or $\phi$), thus the two different distribution forms must be assumed over those two subdomains, as shown in the definition that immediately follows L109. These points have been made more explicit in revision (L128–130). Here, I also realize that the original presentation was slightly out of order: in revision, the paragraph here will be re–structured to a better order (L128–138).

Similar remarks will also be added in the beginning of Sec. 2.5 (L156–161) to suggest in what case we further need to introduced different distribution forms also depending on the sign of $y$. Again, note a slight disorder in presentation here, which has been amended in revision (L156–163).

Section 3: Visualization of the distribution: here, I assume that the Reviewer is asking for a visualization of the distribution defined by Eq. (3.1). However, this is simply a two–dimensional Gaussian distribution, whose form must be widely known. In fact, the use of the multi–dimensional Gaussian distribution is fairly common in the assumed–PDF literature (*e.g.*, Golaz *et al.* 2002, Larson and Golaz 2005). For this reason, instead of visualizations, those references have been simply added in revision (L187).

L195-196 (L218–219 in revision): Choice of $\sigma$: the weights, $\sigma_l$, are chosen as means and variances throughout the paper, because these are the simplest quantities required as outputs. This basic point has been remarked in an earlier part of the manuscript in revision in association with Eq. (2.1.7: L85–87).

L255 (L288 in revision): Choice of $\sigma$: please refer to my response to L195–196 just above.

L293-294 (L316–317 in revision): A possible solution to improve the solution with an assumed PDF: the reason for the failure to capture the basic evolution of the distribution is discussed in the paragraph (L295–300: L318–323 in revision) that immediately follows. Alternative distribution forms are considered in Secs. 4.2 and 4.3 in order to overcome this defect, as clearly stated in the beginning of each section (L305, L343: L328, L367 in revison).

L308 (L331 in revision): An integral over an infinite domain at each time step, as required, is substantially more expensive numerically than just predicting a few PDF parameters. Obviously a sum of far more than few points is required to obtain an integral with an acceptable accuracy. This computation cost, to be performed at every time step, is much more substantial compared to the cost for simply integrating only few parameters in time, as the present method is designed to do. This elaboration has been added in revision (L331–333).

L311 (L335 in revision): "Minor disadvantage": based on the way that the Reviewer is asking the question, it is clear that the Reviewer understands a "disadvantage" that "a distribution can spread to $x < 0$": »the solution covers unphysical values«. Yet, this is only a "minor disadvantage" for the reason explained by the sentence that immediately follows (L312–313).

L348 (L372 in revision): Purpose of section 4.3: the purpose of Sec. 4.3 is to seek "an alternative possibility" *to Sec. 4.2*. In revision, this additional phrase has been added for a further clarity. This method was actually formulated explicitly, but not presented in detail, because as remarked, no substantial difference from the case with Sec. 4.2 was found. I believe it worthwhile to remark on this unsuccessful alternative attempt, even just briefly, because it is a very natural choice to try.

L540: »This study applies the assumed-PDF approach to dynamical systems for which it is possible to compute the solution of the Liouville equation, but the method generally fails to reproduce the exact solution. What would be the appropriate procedure for cases where the exact solution is not available?«: I do not know the answer to this question. However, it seems to me that it is more constructive to seek better assumed–PDF forms that works better for simple dynamical systems. Then, a similar approach can also be applied to more complex systems, where the direct verifications by the Liouville equation is not feasible.

L566-567 (L598 in revision): The sentence redundant?: The sentence of concern has been removed in revision.

**Technical corrections:**

L70 (L92 in revision): The superscript $+$ in $\lambda_l$ in the second integral: thank you for pointing out a typo here. The superscript $+$ here has been corrected to $-$ in revision.

Additional technical corrections suggested by the present Reviewer for L134, Caption in Figs. 3 and 4, L481, L527, L530, L532, L535, and L583 have been performed in revision.

**Response to the Reviewer RC2**

Please refer to the General Remarks and the Summary in the beginning of the file before reading to the more specific response to the Reviewer RC2 here.

I much appreciate the careful balanced thoughts on this manuscript by the present Reviewer.

I much appreciate a positive evaluation by the present Reviewer stating that »the manuscript provides a careful and well-done analytical analysis of the Liouville equations and the assumed Pdf approach.« Yet, at the same time, the Reviewer expresses a strong reservation, remarking that it »likely of rather limited practical benefit as also noted by the author.« Unfortunately, I do not understand what the Reviewer means by »limited practical benefit«. Although the Reviewer says that I also note it, the phrase itself is not of mine.

In response, I emphasize the fact that a very solid analysis is performed in the present study by directly comparing the assumed–PDF results with direct numerical results with the Liouville equation by taking some simple dynamical systems. Providing a general, robust formulation that enables this kind of comparisons is a real originality of the present study: this is not possible with the current existing assumed–PDF schemes, because these are formulated only in a case–by–case manner with only specific applications in mind. Thus the performance of those schemes cannot be tested by simple dynamical systems. See: L560–566 in revision

The robustness of the proposed formulation is discussed extensively in YLP (submitted to ACP), which is also currently available online. The goal of the present manuscript is, as clearly acknowledged by the present Reviewer, to test this formulation for more advanced cases (L23–34).

One may judge that the obtained results are not quite promising, and even a failure. However, one must also count on the basic fact that the assumed PDF attempts something almost impossible: to perform an accurate prediction of a distribution only by using a limited number of parameters. For this reason, one should consider the obtained results are important demonstration of the fundamental difficulties with the assumed–PDF methods in general, not only with a particular approach adopted herein (L575–584).

All these points have been more extensively discussed in the final manuscript.

-Abstract and Conclusion: Here, the Reviewer objects my interpretation that the tendency for vanishing variance with the present approach is "likely a common cause of collapse in variance found in ensemble-based data assimilation". I do not know in what sense »This is not founded here and partly misleading«: it is clearly a common feature that we can identify both with the ensemble–based data assimilation and the present study. Here, I am afraid, the Reviewer slightly misunderstands the standard procedures with ensemble–based data assimilation: as the present Reviewer correctly points out, indeed, they typically »introduce spread/variance through perturbations to the observations and the model« (*cf.*, L582–584: L612–614 in revision). However, this procedure is necessary, because otherwise, the variance would collapse (*cf.*, L574–581: L604–608 in revision), as found in the present study. Somehow the present Reviewer misses this latter point. Please refer to L602–607 of the manuscript in revision.

- Analysis: I am glad to know that the present Reviewer finds that »the analysis is accurately done«.

-The present Reviewer clearly acknowledges that the present manuscript is an extension of another manuscript currently available as a »Technical Memorandum« in EGUsphere (submitted to ACP). The Reviewer states explicitly that »I liked the example/solution part«. Nevertheless, also adds »too long without benefit«, however, with no specific reason provided. More specifically, the Reviewer suggest to drop Secs. 5.2 and 5.3: this suggestion is just odd, because these two subsections pursue the alternative possibilities that partially overcome the defects found with the first model considered in Sec. 5.1. Inclusion of these two subsections is crucial for this reason.

**Typos:**

-l12 (l14 in revision) "as in values themselves": has been modified to "as frequency distributions of variables at a single macroscopic point"

Further typos at l24, l46, and l224, and l287 have also been corrected, as well as the errors in the labeling of Figures 7 and 8.